# Optimal Embedding Guided Negative Sample Generation for Knowledge Graph Link Prediction

**Makoto Takamoto**                                                                          *makoto.takamoto@neclab.eu*
*NEC Laboratories Europe, Heidelberg, Germany*

**Daniel Oõro-Rubio**                                                                        *daniel.onoro@neclab.eu*
*NEC Laboratories Europe, Heidelberg, Germany*

**Wiem Ben Rim**
*University College London, London, UK*

**Takashi Maruyama**
*NEC Laboratories Europe, Heidelberg, Germany*

**Bhushan Kotnis**
*Coresystems AG, Zurich, Switzerland*

**Reviewed on OpenReview:** *https://openreview.net/forum?id=B4SyciDyIh*

## Abstract

Knowledge graph embedding (KGE) models encode the structural information of knowledge graphs to predicting new links. Effective training of these models requires distinguishing between positive and negative samples with high precision. Although prior research has shown that improving the quality of negative samples can significantly enhance model accuracy, identifying high-quality negative samples remains a challenging problem. This paper theoretically investigates the condition under which negative samples lead to optimal KG embedding and identifies a sufficient condition for an effective negative sample distribution. Based on this theoretical foundation, we propose **E**mbedding **MU**tation (EMU), a novel framework that *generates* negative samples satisfying this condition, in contrast to conventional methods that focus on *identifying* challenging negative samples within the training data. Importantly, the simplicity of EMU ensures seamless integration with existing KGE models and negative sampling methods. To evaluate its efficacy, we conducted comprehensive experiments across multiple datasets. The results consistently demonstrate significant improvements in link prediction performance across various KGE models and negative sampling methods. Notably, EMU enables performance improvements comparable to those achieved by models with embedding dimension five times larger. An implementation of the method and experiments are available at `https://github.com/nec-research/EMU-KG`.

## 1 Introduction

Knowledge Graphs (KGs) are graph databases, consisting of a collection of facts about real-world entities that are represented in the form of (head, relation, tail)-triplets. With their logical structure reflecting human knowledge, KGs have proven themselves to be a crucial component of many intelligent systems that tackle complex tasks, such as question answering (Huang et al., 2019), recommender systems (Guo et al., 2022),

information extraction (Gashteovski et al., 2020), machine reading (Weissenborn et al., 2018), and natural language processing, such as language modeling (Yang & Mitchell, 2017; Logan et al., 2019), entity linking (Radhakrishnan et al., 2018), and question answering (Saxena et al., 2022). Popular KGs such as Freebase (Bollacker et al., 2008), YAGO (Suchanek et al., 2007), and WordNet (Miller, 1995) have been instrumental in driving advancements in both academic research and industrial applications.

One of the major challenges that KGs face is their incompleteness; there may be numerous factually correct relations between entities in the graph that are not covered. To address this issue, the task of link prediction has emerged as a fundamental research topic, aimed at filling in the missing links between entities in the graph. Among the various approaches to predicting these missing links, Knowledge Graph Embedding (KGE) methods have proven to be particularly effective. KGE methods encode entities and relations information into a low-dimensional embedding vector space, thus enabling link prediction using neural networks (Bordes et al., 2013; Yang et al., 2015; Trouillon et al., 2016; Sun et al., 2019; Zhang et al., 2020; Abboud et al., 2020; Zhu et al., 2021; Tran & Takasu, 2022).

Various methods have been developed to improve the accuracy of KGE predictions. For instance, Ruffinelli et al. (2020) showed that using contrastive learning improves the model's prediction accuracy, irrespective of the embedding models used. However, to effectively train a model with contrastive learning, it is essential to prepare hard-negative samples that are sufficiently challenging for the model to avoid penalizing true triplets. Although there has been a significant amount of research into effective negative sampling methods (Bordes et al., 2013; Sun et al., 2019; Zhang et al., 2019; Ahrabian et al., 2020; Zhang et al., 2021; Lin et al., 2023; Yao et al., 2023), finding a powerful yet efficient negative sampling method remains an open problem in the research community.

In this paper, we conduct a theoretical investigation into the conditions under which negative samples contribute to optimal embedding in knowledge graph embedding (KGE) models and identify a sufficient condition that the negative sample distribution must satisfy. Building on this theoretical insight, we propose **E**mbedding **MU**tation (EMU), a novel approach for the *generation* of negative samples tailored to the KGE link prediction task. Unlike conventional methods that focus on *identifying* informative negative samples within the training dataset, EMU generates challenging negative samples for training triples by mutating their embedding vectors with components extracted from the target positive embedding vector. By manipulating the components of embedding vectors, EMU efficiently generates negative samples that satisfy the condition for optimal embedding in KGE link prediction tasks. The simplicity of EMU facilitates seamless integration with existing KGE models and any negative sampling strategies. Through extensive experiments across various models and datasets, we demonstrate that EMU consistently delivers substantial performance improvements, highlighting its potential as a robust tool for link prediction. Notably, our experiments reveal that EMU enables models to achieve comparable performance to those with embedding dimensions five times larger, thereby reducing computational complexity.

In summary, our contributions are as follows:

- We theoretically derive and identify a condition for the negative sample distribution that leads to optimal KGE for link prediction tasks.

- We introduce EMU, a novel negative sample generation method that satisfies the identified condition. EMU is compatible with existing KGE models and negative sampling methods, achieving performance comparable to models with significantly larger embedding dimensions.

- We perform comprehensive experiments to validate EMU, demonstrating consistent performance improvements across diverse KGE models, datasets, and negative sampling strategies.

## 2 Background and Notation

**Background** Link prediction is a task that consists of finding new links among entities in a graph by leveraging existing entities and relations. Given a triple (head, relation, tail), one of the elements is omitted

(e.g., (head, relation, ?)), and the model is required to predict the missing element to form a new valid triple [1]. KGE models have demonstrated their effectiveness for this task by learning to represent the knowledge graph structure in a vector space. During training, KGE methods rely on negative sampling techniques because KGs only contain information about positive links. Negative sampling plays a critical role in embedding learning by proposing samples that represent node pairs known *not* to be connected, contrasting with positive samples, which represent connected node pairs. By incorporating negative samples, KGE models improve their ability to distinguish between positive and negative links, enhancing their predictive performance in link prediction tasks. Various techniques have been proposed to propose high-quality negative samples. One of the most widely used methods is Uniform Sampling (Bordes et al., 2013), which corrupts positive triples by replacing either the head or the tail entity with a uniformly sampled alternative from the knowledge graph. However, this approach has notable limitations, as it often produces uninformative samples, leading to limited performance gains due to the potential risk treating unknown positive samples in the KGs as negatives. To address these shortcomings, alternative negative sampling methods have been developed to propose more challenging, or "hard," negative samples, e.g., (Ahrabian et al., 2020).

**Notation** We define a triplet as: $x = (\mathbf{h}, \mathbf{r}, \mathbf{t})$ where $(\mathbf{h}, \mathbf{r}, \mathbf{t})$ denote (head, relation, and tail). These triplets often consists of discrete concepts, such as ('Joe Biden', 'president of', 'USA'), or ('Tokyo', 'capital of', 'Japan'). To facilitate processing by machine learning models, they are usually mapped onto a continuous, low-dimensional latent space $\mathbf{z} \in \mathbb{R}^d$, where $d$ is the dimensionality of the latent space. The mapping is carried out using an embedding function $G$. For instance, to embedding of the head entity is given by $\mathbf{z}^h = G(h|\theta^h)$, where $\theta^h$ is the weight parameters of the embedding model. The feasibility of a triplet is then evaluated using a scoring function $S(z) = s$, where $z = (\mathbf{z}^h, \mathbf{z}^r, \mathbf{z}^t)$ is the latent representation of the input triple and $s$ is the computed score. The scoring function $S$ varies across methods; For example, TransE (Bordes et al., 2013) uses the Euclidean distance, while DistMult (Yang et al., 2015) uses the dot-product.

Training involves minimizing a contrastive loss function that leverages the score of the positive sample $s^+$ from the true triplet and the scores of negative samples $\{s_0, s_1, ...\}^-$, which are produced by corrupting the true triplet:

$$\mathcal{L}(s^+, \{s_0, s_1, ...\}^-) \tag{1}$$

The loss function is designed to increase the score of the true triple while decreasing the score of negative samples. Depending on the specific method, this optimization can involve minimizing or maximizing distances (as in TransE) or optimizing similarities (as in DistMult).

## 3 Optimal Embedding for KGE and EMU

In this section, we illustrate how the negative samples generated by EMU lead to near-optimal embedding for KGE link prediction problems. First, we introduce the principle of negative sampling for graph representation learning proposed by (Yang et al., 2020), and extend it to the KGE link prediction problems. Next, we provide a comprehensive description of EMU. Finally, we demonstrate that EMU generates negative samples that distribute isotropically around positive samples, which leads to the condition for the principle of negative sampling for KGE link prediction problems.

### 3.1 Optimal Embedding for KGE Representation Learning

Drawing upon the theoretical framework proposed by (Yang et al., 2020), referred to as Y20 henceforth, we posit that the target node $v$ and the positive node $u$, each characterized by their respective embedding vectors, $\mathbf{u}, \mathbf{v}$, are derived from the positive sample distribution: $p_d(u|v)$. An objective function for embedding

---

[1]If either "head" or "tail" is omitted, the task is referred to as "entity prediction"; When the "relation" is omitted, it is denoted as "relation prediction". While this paper mainly discusses the "tail" prediction scenario for simplicity, the proposed method is applicable to other cases as well.

is:

$$J^{(v)} = \mathbb{E}_{v \sim p_d(v)}[\mathbb{E}_{u \sim p_d(u|v)} \log \sigma(\mathbf{u} \cdot \mathbf{v}) + k \ \mathbb{E}_{u' \sim p_n(u'|v)} \log \sigma(-\mathbf{u}' \cdot \mathbf{v})], \tag{2}$$

where $p_n(u|v)$ is the negative sample distribution and $\sigma(\cdot)$ the sigmoid function, and $k$ is a numerical constant; and its corresponding empirical risk for node v is:

$$J_{\mathrm{T}}^{(v)} = \frac{1}{T} \sum_{i=1}^{T} \log \sigma(\mathbf{u}_i \cdot \mathbf{v}) + \frac{1}{T} \sum_{i=1}^{kT} \log \sigma(-\mathbf{u}'_i \cdot \mathbf{v}), \tag{3}$$

where $\{u_1, \cdots, u_T\}$ are sampled from $p_d(u|v)$ and $\{u'_1, \cdots, u'_k\}$ are sampled from $p_n(u|v)$. We set $\theta = [\mathbf{u_0} \cdot \mathbf{v}, \ldots, \mathbf{u_{N-1}} \cdot \mathbf{v}]$ as the parameters to be optimized, where $\{u_0, \cdots, u_{N-1}\}$ are all the $N$ nodes in the graph.

Under the assumption, Y20 derived the following equation, as an empirical realization of the covariance measure proved in Y20:

$$\mathbb{E}[||(\theta_T - \theta^*)_u||^2] = \frac{1}{T} \left( \frac{1}{p_d(u|v)} - 1 + \frac{1}{k p_n(u|v)} - \frac{1}{k} \right). \tag{4}$$

Here, $\theta_T$ and $\theta^*$ are the respective optimal parameters for $J_T^{(v)}$ and $J^{(v)}$. Based on this theoretical finding, Y20 proposed a principle of negative sampling which enables the optimal $J_{\mathrm{T}}^{(v)}$ to converge to the optimal $J^{(v)}$ for graph representation learning. We extend the idea to KG link prediction, assuming a KGE model "DistMult" (Yang et al., 2015) which calculate the scoring function as: $s_{\mathrm{DistMult}} \equiv \sum(\mathbf{z}^h \odot \mathbf{z}^r \odot \mathbf{z}^t)$ where $\odot$ is the element-wise multiplication. The extension is given as:

**Theorem 3.1.** *Assuming DistMult model, an empirical realization of the covariance measure can be given as,*

$$\mathbb{E}[||(\theta_T - \theta^*)_{\mathbf{z^t}}||^2] = \frac{1}{T} \left( \frac{1}{p_d(\mathbf{z}^t|v^{hr})} - 1 + \frac{1}{k p_n(\mathbf{z}^t|v^{hr})} - \frac{1}{k} \right). \tag{5}$$

*where $\mathbf{v}^{hr} = \mathbf{z}^h \odot \mathbf{z}^r$ and $\mathbf{z}^h, \mathbf{z}^r, \mathbf{z}^t$ are the head, relation, and tail embedding vectors, respectively.*

*Proof.* Equation 5 can be derived by substituting $\mathbf{z}^t$ and $\mathbf{z}^h \odot \mathbf{z}^r$ for $\mathbf{u}, \mathbf{v}$ in the proof provided in Y20. $\square$

### 3.2 EMU: Embedding MUtation

EMU is inspired from the gene 'mutation' technique utilized in evolutionary algorithms. In this study, we propose a new non-linear mixing approach that replaces a certain amount of the embedding vector components in the negative sample with the corresponding parts of the true positive vector components. This technique is a simple yet effective means of enhancing the difficulty of negative samples by increasing their similarity to the true positive. Figure 1 provides a two-dimensional visualization of this phenomena.

The formal definition of the EMU technique is presented as follows:

$$\tilde{\mathbf{z}}_{\mathrm{EMU}} = \lambda_{\mathrm{EMU}} \odot \mathbf{z}^+ + (1 - \lambda_{\mathrm{EMU}}) \odot \mathbf{z}^-, \tag{6}$$

where $\lambda_{\mathrm{EMU}} \in \mathbb{R}^d$ is the EMU mixing vector that controls the number of embedding vector components to be mutations, which is denoted as $n_{\mathrm{P}}$. $\lambda_{\mathrm{EMU}}$ is a binary-valued vector whose components are generated through a random sampling process that selects either zero or unity, with the probability of $(1 - n_{\mathrm{P}}/d, n_{\mathrm{P}}/d)$. [2] The symbol $\odot$ denotes element-wise multiplication, and $\mathbf{z}^+$ and $\mathbf{z}^-$ correspond to the positive and negative vectors to be mutated, respectively.

For the application of knowledge-base link prediction, we utilize Equation 6 to create the EMU negative tail sample by substituting $\mathbf{z}^t = (\mathbf{z}^{t,+}, \{\mathbf{z}_0^t, \mathbf{z}_1^t, \cdots\}^-)$. We employ the generated samples as the EMU negative samples.

---

[2]More concretely, the component of the vector $\lambda_{\mathrm{EMU}} \in \mathbb{R}^d$ is composed by $n_{\mathrm{P}}$ unities and $d - n_{\mathrm{P}}$ zeros whose order is randomly determined, e.g., $\{0, 1, 1, 0, 0, 0, \cdots\}$. For simplicity we use the random sampling. The study of the better mutation vector $\lambda_{\mathrm{EMU}}$ is our future work.

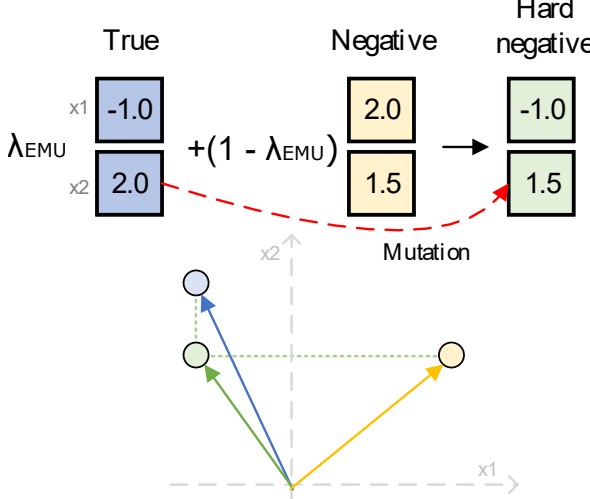

Figure 1: EMU generates a new negative samples with embedding mutation. The figure illustrates a typical example that generate hard negative tails.

### 3.3 Theoretical Consideration on EMU

In the following, we demonstrate that EMU generates negative samples that are isotropically distributed around positive samples, thereby satisfying the condition necessary for achieving optimal embedding. Specifically, we present the following theorems to illustrate that the negative samples produced by EMU exhibit isotropic distribution around the positive samples.

**Theorem 3.2.** *Suppose that* $\mathbf{A}$ *and* $\mathbf{B}$ *are vectors in* $\mathbb{R}^N$ *and* $\mathbf{B}_{\mathrm{EMU}} \equiv \lambda_{\mathrm{EMU}} \odot \mathbf{B}$. *Assuming a sufficiently small standard deviation in the vector component of* $\mathbf{B}$ *with non-zero mean value and sufficiently large* $N$, *the angle formed between* $\mathbf{A}$ *and* $\mathbf{B}_{\mathrm{EMU}}$ *adheres to a Gaussian distribution whose center is the angle between* $\mathbf{A}$ *and* $\mathbf{B}$.

The proof for this theorem can be found in Appendix C. Utilizing the above theorem, we can deduce:

**Theorem 3.3.** *EMU generates negative samples isotropically around the target positive sample.*

*Proof.* According to the definition (6), the deviation vector $\Delta z^\lambda$ can be written as:

$$\Delta z^\lambda \equiv \tilde{\mathbf{z}}_{\mathrm{EMU}} - \mathbf{z}^+ = (\mathbf{1} - \lambda_{\mathrm{EMU}}) \odot \Delta \mathbf{z}, \tag{7}$$

where $\Delta \mathbf{z} \equiv \mathbf{z}^- - \mathbf{z}^+$. Subsequently, we consider the angular distribution of $\{\Delta z_i^\lambda\}_{i=1,\dots k}$. Without loss of generality, we take the angle distribution of $\Delta \mathbf{z}^\lambda$ in terms of $\Delta z_0^\lambda$. The cosine function between $\Delta \mathbf{z}_i^\lambda$ and $\Delta \mathbf{z}_0^\lambda$ can be expressed as:

$$\{\cos \theta_i^\lambda\} \equiv \left\{ \frac{\Delta \mathbf{z}_0^\lambda \cdot \Delta \mathbf{z}_i^\lambda}{|\Delta \mathbf{z}_0^\lambda||\Delta \mathbf{z}_i^\lambda|} \right\}$$
$$= \left\{ \frac{\Delta \mathbf{z}_0^\lambda}{|\Delta \mathbf{z}_0^\lambda||\Delta \mathbf{z}_i^\lambda|} \cdot (\mathbf{1} - \lambda_{\mathrm{EMU,i}}) \odot \Delta \mathbf{z}_i \right\}. \tag{8}$$

In this context, the operator $(\mathbf{1} - \lambda_{\mathrm{EMU},i})\odot$ projects $\Delta \mathbf{z}_i$ onto a lower-dimensional hypersurface, and Theorem 3.2 hence shows $(\mathbf{1} - \lambda_{\mathrm{EMU},i})\odot$ induces the Gaussian distribution in terms of the angle between $\Delta \mathbf{z}_0^\lambda$ and $\Delta \mathbf{z}_i^\lambda$ in the angle-space. This results in isotropic distribution of the EMU samples in the coordinate space, which are isotropically dispersed around the target positive tail sample. □

Before proving the final theorem, we provide the following lemma. In the rest of this subsection, we will consider $\tilde{\mathbf{z}}_{\mathrm{EMU}}, \mathbf{z}^{t,+}, \Delta \mathbf{z}^\lambda$ as independent random variables.

**Lemma 3.4.** *In the context of EMU, $p_{n,\mathrm{EMU}}(\mathbf{z}^t|\mathbf{v}^{hr})$ can be expressed as:*

$$p_{n,\mathrm{EMU}}(\tilde{\mathbf{z}}_{\mathrm{EMU}}|v^{hr}) = A \int d\mathbf{z}^{t,+} \int d\Delta\mathbf{z}^\lambda \left[\delta_D(\tilde{\mathbf{z}}_{\mathrm{EMU}} - \mathbf{z}^{t,+} - \Delta\mathbf{z}^\lambda) f_{\mathrm{iso}}(|\Delta\mathbf{z}^\lambda| \ |\mathbf{z}^{t,+}, \Delta\bar{z}) p_d(\mathbf{z}^{t,+}|\mathbf{v}^{hr})\right] \quad (9)$$

*where $\delta_D$ is the Dirac delta function, $f_{\mathrm{iso}}$ is an isotropic distribution with a typical decaying scalar scale $\Delta\bar{z}$, and $A$ is an appropriate constant.*

The proof for this lemma can be found in Appendix D. To finalize, we can conclude the following claim:

**Theorem 3.5.** *Assuming EMU considers the embedding vector of negative samples to be muted in the neighbor of the considering positive sample, the negative sample distribution by EMU results in near-optimal embedding.*

*Proof.* According to Lemma 3.4, $f_{\mathrm{iso}}$ is a sufficiently rapidly decreasing function when $|\Delta\mathbf{z}^\lambda| > \Delta\bar{z}$. In the following, we approximate $f_{\mathrm{iso}}$ by the Heaviside function: $H(\Delta\bar{z} - |\Delta\mathbf{z}^\lambda|)$, which does not affect the following order-of-magnitude discussion. Concretely, the variation within $|\Delta\mathbf{z}^\lambda| < \Delta\bar{z}$ can be taken into account by properly renormalize the coefficient denoted as $A$ in the following. Then, Equation 9 reduces to:

$$
\begin{aligned}
p_{n,\mathrm{EMU}}(\tilde{\mathbf{z}}_{\mathrm{EMU}}|\mathbf{v}^{hr}) &\simeq A \int d\Delta\mathbf{z}^\lambda \ H(\Delta\bar{z} - |\Delta\mathbf{z}^\lambda|) p_d(\tilde{\mathbf{z}}_{\mathrm{EMU}} - \Delta\mathbf{z}^\lambda|\mathbf{v}^{hr}) \\
&\simeq A \int_0^{\Delta\bar{z}} d|\Delta\mathbf{z}^\lambda| \int_0^{4\pi} d\Omega_{\Delta z^\lambda} p_d(\tilde{\mathbf{z}}_{\mathrm{EMU}} - \Delta\mathbf{z}^\lambda|\mathbf{v}^{hr}) \\
&\simeq A' \ p_d(\tilde{\mathbf{z}}_{\mathrm{EMU}}|\mathbf{v}^{hr}) + \mathcal{O}((\Delta\bar{z}^\lambda)^2),
\end{aligned} \quad (10)
$$

where $A'$ is a constant. To obtain the final line, we performed the Taylor expansion in terms of $|\Delta\mathbf{z}^\lambda|$ and performed the integration in terms of $\Delta z^\lambda$. Then, by substituting the above equation, Equation 5 reduces to:

$$\mathbb{E}[||(\theta_T - \theta^*)_{\mathbf{z}^t}||^2] = \frac{1}{T}\left(\frac{1}{p_d(\mathbf{z}^t|\mathbf{v}^{hr})}\left[1 + \frac{1 - \mathcal{O}((\Delta\bar{z}^2)}{kA'}\right] - 1 - \frac{1}{k}\right). \quad (11)$$

As in Y20, this indicates that the order of magnitude of deviation is only negatively related to $p_d$ for the case of EMU. $\qquad\square$

Theorem 3.5 suggests that the distribution of negative samples generated by EMU closely aligns with Equation 5 as stated in Theorem 3.1, indicating that EMU produces near-optimal negative sample distribution. Note that Theorem 3.5 also shows that the near-optimal embedding can be achieved when negative samples are distributed according to a rapidly decreasing isotropic function, $f_{\mathrm{iso}}$, around the target positive samples. EMU facilitates the easy generation of such negative samples. Note that we have not explicitly used the fact that the objective function depends on $s_{\mathrm{DistMult}}$. The logic in the above proof and Y20 can be applied to a more generic form of the embedding, $f(\mathbf{z^h}, \mathbf{z^r}, \mathbf{z^t})$ where $f$ is a scalar function that takes $\mathbf{z^h}, \mathbf{z^r}, \mathbf{z^t}$ as inputs. This formulation encompasses models such as the DistMult and neural networks.

### 3.4 Overall Loss Terms

Inspired by knowledge distillation (Hinton et al., 2015), we combine the EMU loss function with the usual loss function without EMU, enabling the model to learn from the vanilla negative samplings (i.e., sampled using the existing methods) as well. The overall loss function is expressed as:

$$\mathcal{L} = \mathcal{L}(s^+, \{s_{0,\mathrm{EMU}}, s_{1,\mathrm{EMU}}, \cdots\}^-; \bar{y}) + \alpha\mathcal{L}(s^+, \{s_0, s_1, \cdots\}^-; \hat{y}), \quad (12)$$

where $\mathcal{L}$ is a contrastive loss function, $\hat{y}$ is the one-hot label, and $\bar{y}$ is a generalized label by such as a label-smoothing. The numerical coefficient $\alpha$ is utilized for weight balancing between the losses.

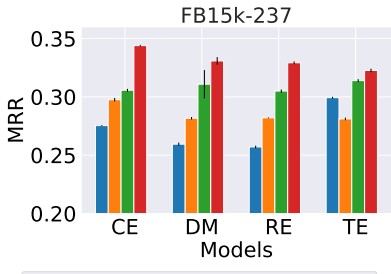 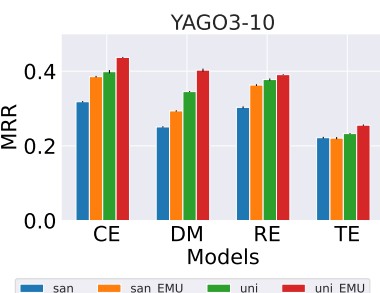 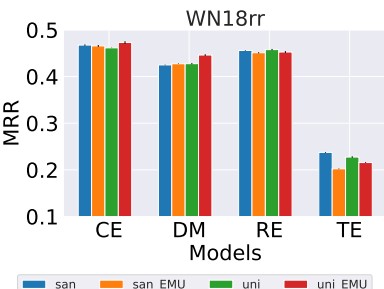

Figure 2: MRR for the datasets: FB15k-237, YAGO3-10, and WN18RR. The blue, orange, green, and red colored bars mean the result of using the following negative sampling methods: "SAN", "SAN with EMU", "uniform", and "uniform with EMU", respectively.

## 4 Emperical Validation

In this section, we perform an experimental evaluation of EMU for the link prediction problem to validate the effectiveness of EMU as shown in section 3. To ensure a thorough evaluation, we chose commonly used KG embedding models (ComplEX (Trouillon et al., 2016; Lacroix et al., 2018), DistMult (Yang et al., 2015; Salehi et al., 2018), TransE (Bordes et al., 2013), RotatE (Sun et al., 2019)), HAKE (Zhang et al., 2020), and NBFNet (Zhu et al., 2021) to test with EMU. Although we used the DistMult model in section 3, the above models enable to assess the effectiveness of EMU to more generic form of operation for triplets than the inner-product type one. Furthermore, we evaluate them on three widely used knowledge graphs, namely FB15k-237 (Toutanova & Chen, 2015), WN18RR (Dettmers et al., 2018), and YAGO3-10 (Mahdisoltani et al., 2013) whose detailed statistics are provided in Appendix F.

### 4.1 Experimental Setup

In order to enable a fair comparison between the different models and to ensure that all methods are evaluated under the same conditions, we implemented all the methods [3]. Among all existing baselines, we consider vanilla uniform negative sampling (Bordes et al., 2013), SAN (Ahrabian et al., 2020), and NScaching (Zhang et al., 2019) as the most relevant to compare our work against. A detailed experimental setup is provided in Appendix G. For the loss function in Equation 12, we consider the cross-entropy loss function with a slightly modified label-smoothing as $\bar{y}$, which is described in Appendix E. [4]

### 4.2 Results

This section provides a summary and discussion of the obtained results.

Figure 2 illustrates the quantitative results in Table 8, displaying three plots for the MRR results obtained for the FB15k-237, YAGO3-10, and WN18rr datasets. The results of HAKE model are provided in Table 9 and the results with NScaching are provided in Table 10. Each plot includes four groups of column bars, representing the results for ComplEX (Trouillon et al., 2016; Lacroix et al., 2018) (CE), DistMult (Yang et al., 2015; Dettmers et al., 2018; Salehi et al., 2018) (DM), RotatE (Sun et al., 2019) (RE), TransE (Bordes et al., 2013) (TE). The columns are distinguished by colors that correspond to the results obtained from running the SAN (in blue), SAN_EMU (i.e.: SAN negative sampling method with EMU, in orange), uniform sampling (green), and uni_EMU (i.e.: simple uniform negative sampling with EMU, in red). The findings demonstrate that, in most cases, employing EMU significantly improves the scores across all embedding models. An exception is observed with the WN18rr dataset, which is further analyzed and theoretically examined in subsection 4.4.

---

[3]The code to replicate our experiments can be found: `https://anonymous.4open.science/r/EMU-KG-6E58`.

[4]Note that the assumed loss function in section 3, that is, the sigmoid-type loss function Equation 3, is related to but different from Equation 12 which is one of the recent-popular loss functions (Ruffinelli et al., 2020). The following experimental results also indicates the wider applicability of EMU than assumed in section 3.

| Dataset | Model | Case | Parameter: (d, n) | MRR | HITS@10 |
|---|---|---|---|---|---|
| FB15k-237 | DistMult | w/t EMU | (200, 64) | 0.314 | 0.500 |
| | | w/t EMU | (1000, 64) | 0.327 | **0.519** |
| | | EMU | (200, 64) | **0.329** | 0.514 |
| | HAKE | w/t EMU | (200, 256) | 0.175 | 0.315 |
| | | w/t EMU | (1000, 256) | 0.308[a] | 0.493[a] |
| | | EMU | (200, 256) | **0.311** | **0.501** |
| YAGO3-10 | DistMult | w/t EMU | (100, 256) | 0.345 | 0.538 |
| | | w/t EMU | (1000, 256) | 0.393 | 0.595 |
| | | EMU | (100, 256) | **0.403** | **0.601** |
| | HAKE | w/t EMU | (500, 256) | 0.452 | 0.651 |
| | | w/t EMU | (1000, 256) | 0.482 | 0.665 |
| | | EMU | (500, 256) | **0.499**[a] | **0.687**[a] |

Table 1: Embedding dimension efficacy study results on FB15K-237 and YAGO3-10. (d, n) denote the embedding dimension and the negative sample number. "w/t EMU" denotes the model trained without EMU but with uniform sampling. The best performance is written in bold font and the second best performance is written with underline.

[a] The reported values are obtained by our own training of HAKE model with utilizing the official repository (Zhang et al., 2020).

## 4.3 Embedding Dimension Efficacy Study

In this subsection, we study the embedding dimension efficacy of EMU using FB15K-237 and YAGO3-10. For this study, we consider DistMult and HAKE models as classical and recent representative KGE models. To show EMU efficacy, we performed the training of the models with large embedding dimension without EMU in comparison with the one with small embedding dimension with EMU. The result is provided in Table 1, which shows that EMU enables to achieve the model performance comparable to five times larger embedding dimension case. In particular, HAKE results show a significant performance gain, indicating that EMU enables the recent sophisticate model with smaller size but keeping the prediction performance.

## 4.4 Mutation effect

In this subsection, we analyze and discuss the mutation effect in terms of embedding similarity. We use DistMult as a reference model and train it on FB15k-237 and WN18RR datasets. We visualize the embedding vector of the negative tail obtained with EMU to compare it with other negative sampling strategies, i.e., uniform random sampling and SAN negative sampling.

Figure 3 shows the cosine similarity of negative samples provided by the three strategies. The similarity distributions of negative samples produced by the *uniform* and *SAN* methods are quite low, resulting in "easy" negative samples. In contrast, the negative samples generated by EMU exhibits a much larger similarity, indicating that EMU generates harder negative samples than the other methods as shown in Proposition B.1.

Figure 4 depicts the distribution of true-tails and negative tails for two different datasets, namely FB15k-237 and WN18rr, by plotting the first and second PCA components. The left panel of each figure shows the distribution when negative tails are uniformly sampled, while the right panel depicts the distribution using EMU negative-tails. In the Figure 4, the distribution around a true tail is anisotropic for uniform-negative sampling, while EMU negative-tails show a rapidly decreasing isotropic distribution as shown in Theorem 3.3, which validates our assumption of $f_{\mathrm{iso}}$ by the Heaviside function. Moreover, the distributions for FB15k-237 are quite varied in comparison to those of WN18RR. This could be an explanation for the higher performance gain when using EMU for FB15k-237 (see Table 8)[5].

---

[5]The isotropic distribution of embedded node vectors observed in the WN18RR dataset is consistent with findings from previous studies (Sadeghi et al., 2021), highlighting the influence of node degree and centrality measures on node embeddings (Sardina et al., 2024; He et al., 2022a; Shomer et al., 2023). Furthermore, we note that the performance improvements of

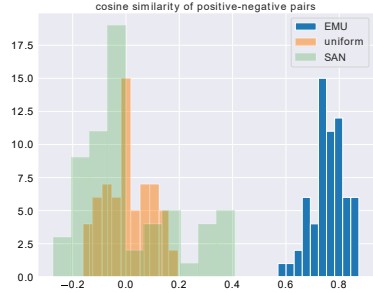

Figure 3: Cosine similarity between positive and negative sample pair for DistMult trained on FB15k-237 dataset. The used negative samples are: uniform, EMU, and SAN. The larger, the more similar.

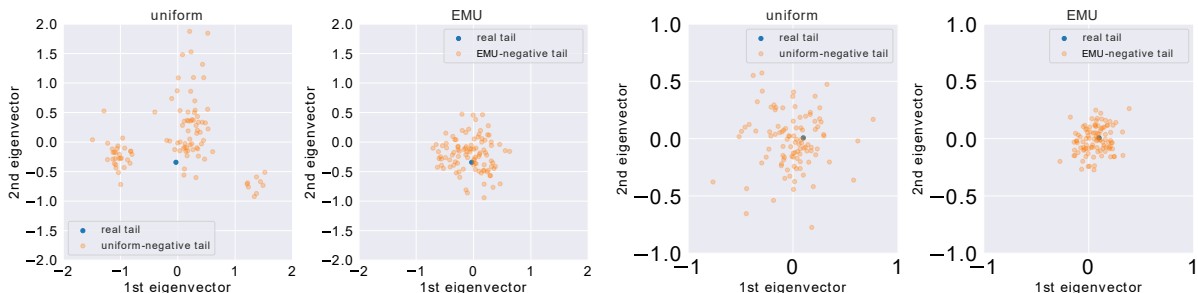

Figure 4: Results of the analysis of EMU of DistMult model trained on FB15k-237 (Left) and WN18rr (Right) dataset. Left: The distribution of real-tail and uniformly-sampled negative-tail in terms of the 1st and 2nd PCA components. Right: The distribution of real-tail and EMU negative-tail in terms of the 1st and 2nd PCA components.

| Model | Method | FB15K-237 | | WN18RR | | YAGO3-10 | |
|---|---|---|---|---|---|---|---|
| | | MRR | HITS@10 | MRR | HITS@10 | MRR | HITS@10 |
| ComplEX | uni_EMU | **0.344** | **0.532** | **0.473** | **0.547** | **0.437** | **0.638** |
| (Lacroix et al., 2018) | uni_Mixup | 0.324 | 0.517 | 0.470 | **0.547** | 0.418 | 0.605 |
| DistMult | uni_EMU | **0.332** | **0.513** | **0.446** | **0.523** | **0.403** | **0.601** |
| (Yang et al., 2015) | uni_Mixup | 0.319 | 0.507 | 0.441 | 0.517 | 0.402 | 0.597 |
| RotatE | uni_EMU | **0.329** | **0.514** | 0.453 | 0.525 | **0.365** | **0.555** |
| (Sun et al., 2019) | uni_Mixup | 0.281 | 0.454 | 0.278 | 0.428 | 0.174 | 0.318 |
| TransE | uni_EMU | **0.323** | **0.503** | 0.216 | 0.493 | **0.255** | **0.436** |
| (Bordes et al., 2013) | uni_Mixup | 0.269 | 0.421 | 0.050 | 0.139 | 0.063 | 0.102 |

Table 2: MRR and Hit@10 of the results on FB15K-237, WN18RR, and YAGO3-10 datasets. "uni_EMU" means the uniform negative sampling with EMU and "uni_Mixup" means the uniform negative sampling with Mixup. The shown results are the average of three trials of the randomly determined initial weights.

### 4.5 Comparison to Mixup

There is a well-known approach called Mixup, which shares a similar philosophy with the EMU but is based on a different theoretical background (see also section 5). To compare their performance, we replaced

---

DistMult and ComplEx remain significant, with gains ranging from 5% to 10%. This improvement may be attributed to the ability of DistMult-type models to better capture low-degree entities compared to TransE (Rossi & Matinata, 2020).

the Embedding Mutation step with Mixup and present the results in Table 2 [6]. The findings consistently demonstrate that EMU outperforms Mixup. We hypothesize that the linear nature of Mixup-generated examples limits the magnitude of gradients while preserving their direction, thereby restricting its effectiveness. In contrast, EMU overcomes this limitation by generating updates that can explore multiple directions, thereby enhancing model training

### 4.6 Application to NBFNet

In this section, we apply EMU to Neural Bellman-Ford Networks (NBFNet) (Zhu et al., 2021), one of the state-of-the-art (SOTA) models for knowledge graph tasks. The results on the FB15K-237 dataset are presented in Table 3, demonstrating that EMU remains effective even when applied to the latest models, achieving performance competitive with the current SOTA methods[7].

| Model | FB15K-237 | | | |
| --- | --- | --- | --- | --- |
| | MRR | HITS@1 | HITS@3 | HITS@10 |
| NBFNet (reported) | 0.415 | 0.321 | 0.454 | 0.599 |
| NBFNet (our experiments) | $0.414 \pm 0.003$ | $0.321 \pm 0.003$ | $0.453 \pm 0.003$ | $0.596 \pm 0.002$ |
| NBFNet w.t. EMU | $\mathbf{0.419 \pm 0.002}$ | $\mathbf{0.326 \pm 0.002}$ | $\mathbf{0.460 \pm 0.003}$ | $\mathbf{0.601 \pm 0.002}$ |

Table 3: MRR, Hit@1, Hit@3, and Hit@10 of the results of NBFNet on FB15K-237 dataset. The results are the average with the standard deviation of three trials of the randomly determined initial weights.

## 5 Related Work

**KGE Models**   KGE models such as TransE(Bordes et al., 2013), DistMult (Yang et al., 2015; Dettmers et al., 2018; Salehi et al., 2018), ConvE (Dettmers et al., 2018), ComplEX (Trouillon et al., 2016; Lacroix et al., 2018), RotatE (Sun et al., 2019) are commonly used when solving the knowledge base completion task. Each model implements a scoring function mapping a given triple to a real-valued number. These models also differ in the embedding spaces used to learn the latent embedding, for instance RotatE (Sun et al., 2019) utilizes the complex vector space. The new KGE model investigation is still actively conducted Zhang et al. (2020); Abboud et al. (2020); Zhu et al. (2021); Tran & Takasu (2022); Zhang et al. (2023); Zhu et al. (2024); Zhou et al. (2024) and more comprehensive review can be found in Ge et al. (2024).

**Negative sampling**   While training a KGE model for the link prediction task, it is essential to sample high-quality negative data points adequately from the graph. Poor quality negative samples can hinder the performance of KGE models by failing to guide the model during training. With this in mind, many approaches were proposed for generating better-quality negative samples, i.e., hard negatives. The earliest sampling method is Uniform Sampling (Bordes et al., 2013). Another commonly used method relies on Bernoulli Sampling where the replacement of the heads or tails of the triples follows the Bernoulli distribution. (Wang et al., 2014). Newer methods that are based on Generative Adversarial Networks (GAN) are also used such as KBGAN (Cai & Wang, 2018) and IGAN (Vignaud, 2021) where the generator is adversarially trained for the purpose of providing better quality negative samples where a KGE model is used as the discriminator. Building on this NScasching (Zhang et al., 2019) proposed a distilled version of GAN-based methods by creating custom clusters of candidates entities used for the negative samples. Structure Aware Negative Sampling (SANS) (Ahrabian et al., 2020) leverages the graph structure in the KG by selecting negative samples from a node's k-hop neighborhood. In addition, the subject continues to be actively studied (Zhang et al., 2021; Islam et al., 2022; Xu et al., 2022; Lin et al., 2023; Yao et al., 2023; Chen et al., 2023; Qiao et al., 2023) . Unlike the prior work mentioned above, EMU generates hard negative samples, distinct from traditional approaches aimed at identifying more difficult negative samples. Furthermore, an additional

---

[6]Note that this procedure is already different from the original implementation of Mixup which corrupts the original data, not that of embedding dimension.

[7]While the performance improvement is relatively smaller compared to other models, this may be attributed to the additional structure of NBFNet, specifically the Bellman-Ford iteration, which differs from the architectures used in other experiments.

benefit of EMU is its compatibility with any of the above negative sampling methods, allowing for seamless integration.

## 6 Discussion and Conclusion

In the present study, we theoretically identified a condition of the nice negative sample distribution leading to a near-optimal embedding of KGE and identified a sufficient condition for near optimal embedding. Based on the condition, we proposed EMU which aims to generate the negative samples following the condition easily and efficiently. Our comprehensive experimental findings also demonstrate that EMU consistently outperforms all the baseline negative sampling methods, including uniform sampling, SAN, and NSCaching in almost all the KGE models and datasets. We also observed that EMU's efficacy was largely invariant across embedding models and datasets. Moreover, the experiments shows that EMU enables to achieve comparable performance to models with embedding dimensions that are five times larger. Our analysis showed that EMU generates negative samples that are closer to true samples in terms of cosine-similarity, and that the generated samples exhibit a more isotropic distribution around the true sample in the embedding space compared to other methods, which is a part of the condition for the near optimal embedding for KGE. Although EMU involves tuning a few hyper-parameters, we observed that its performance is not heavily reliant on them (refer to Appendix K). While EMU was developed for KGE tasks, its simple structure enables its application to other tasks, such as graph node classification and representation learning. Exploring these applications remains a promising direction for future work.

## Acknowledgements

The authors would like to thank their colleagues at NEC Laboratories Europe, in particular Dr. Kiril Gashteovski, Dr. Carolin Lawrence, and Professor Mathias Niepert, for their valuable feedback and encouraging support throughout this work.

## 7 Limitations

EMU scope is restricted to KG missing link prediction model trained using the cross-entropy loss function with negative samples. It cannot be applied to neither 1-VS-ALL method nor the other loss functions for the moment.

## Potential Broader Impact and Ethical Aspects

This paper presents work whose goal is to advance the field of Machine Learning, in particular, knowledge-graph link prediction. There are many potential societal consequences of our work in the far future due to the generic nature of pure science, none which we feel must be specifically highlighted here. While we do not foresee a substantial ethical concern in our proposed strategies, there may be a side effect resulting from the feature mutation. It is thus important to monitor and evaluate potential bias that may arise during the model training process. Note that we utilized only publicly available KG datasets, and thus, there is no concerns regarding compliance with the General Data Protection Regulation (GDPR) law[8]

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

## A    Additional Related Work

**Model Regularization Methods for Classification Tasks**    To obtain a good representation of the embedding vector of a machine learning model, it is common to consider regularization methods. In particular, there have been several regularization techniques for a better generalization power in the case of the cross entropy loss function. MIXUP (Zhang et al., 2018) is one of the most popular and powerful regularization methods, originally developed for image and speech processing. This method generates new training samples by convexly mixing two different training data during the training, resulting in a network with a better generalization because of Vicinal Risk Minimization (Chapelle et al., 2000). Consequently MIXUP has gained popularity in computer vision (Liu et al., 2021; He et al., 2022b; Wang et al., 2021) and voice recognition (Meng et al., 2021; Fang et al., 2022), among other fields (Tolstikhin et al., 2021; Kalantidis et al., 2020; Roy et al., 2022; Che et al., 2022). CUTMIX Yun et al. (2019) is a variant of MIXUP that combine two input images as MIXUP but by cutting and pasting patches among images. Our Feature Mutation shares a similar philosophy but the crucial difference is that feature mutation combines positive and negative tails in "feature" space that has not yet been tried in any existing work as far as we know.

Label Smoothing (Szegedy et al., 2016; Müller et al., 2019) is also known as a very effective regularization method when combined with cross entropy loss. Label Smoothing prevents overconfident predictions from the model by artificially reducing the true labels to be less than unity.

## B    Further Property of EMU

The following proposition describes that EMU generates more difficult negative samples than the original ones.

**Proposition B.1.** *The generated negative samples generated by EMU are always closer to the positive samples than the original negative samples.*

*Proof.* The Euclidean distance between a positive sample $\mathbf{z}^+ = \{z_i^+\}_{i=1,\ldots,d}$ and a vanilla negative sample $\mathbf{z}^- = \{z_i^-\}_{i=1,\ldots,d}$ can be written as:

$$d_{\mathrm{PN}} = \sqrt{\sum_{i=1}^{d}(\mathbf{z}_i^+ - \mathbf{z}_i^-)^2}. \tag{13}$$

On the other hand, the distance between a positive sample and a negative sample generated using EMU is

$$
\begin{aligned}
d_{\mathrm{EMU}} &= \sqrt{\sum_{i=1}^{d}(\mathbf{z}_i^+ - \mathbf{z}_i^{\mathrm{EMU}})^2} \\
&= \sqrt{\sum_{i=1}^{d}\left(\mathbf{z}_i^+ - \{\lambda_i \mathbf{z}_i^+ + (1-\lambda_i)\mathbf{z}_i^-\}\right)^2} \\
&= \sqrt{\sum_{i=n_{\mathrm{T}}+1}^{d}(\mathbf{z}_i^+ - \mathbf{z}_i^-)^2} \leq d_{\mathrm{PN}},
\end{aligned}
\tag{14}
$$

where in the last line, we assume the first $n_{\mathrm{P}}$ components in $\lambda_{\mathrm{EMU}}$ is unity and the others are zero: $\lambda_{\mathrm{EMU}} = \{1, 1, \cdots, 1, 0, \cdots, 0\}$, without loss of generality.

The above equations show that EMU enables to generate hard negative samples than the original ones. $\qquad\square$

## C    Proof of Theorem 3.2

In the following, we provide the proof of Theorem 3.2.

*Proof.* For simplicity, we assume the first axis of the coordinate is aligned with vector $\mathbf{A}$: $\mathbf{A} \equiv a_1 \mathbf{e}_1$ and we also represent $\mathbf{B}$ as $\mathbf{B} = \{b_i\}_{i=1,\dots,N} = \{\bar{b} + \Delta b_i\}_{i=1,\dots,N}$ where $\bar{b}$ is the mean-value of the component of the vector $\mathbf{B}$. Then, the angle between $\mathbf{A}$ and $\mathbf{B}$ can be written as:

$$\cos\theta_0 \equiv \frac{\mathbf{A} \cdot \mathbf{B}}{|\mathbf{A}||\mathbf{B}|} = \frac{b_1}{\sqrt{\sum_{i=1}^{N}(\bar{b} + \Delta b_i)^2}}$$
$$= \frac{b_1}{\bar{b}\sqrt{N}} \left[1 + \frac{1}{N}\sum_{i=1}^{N}\left(\frac{\Delta b_i}{\bar{b}}\right)^2\right]^{-1/2}. \tag{15}$$

Likewise, the angle between $\mathbf{A}$ and $\mathbf{B}$ can be written as:

$$\cos\theta_{\mathrm{EMU}} \equiv \frac{\mathbf{A} \cdot \mathbf{B}_{\mathrm{EMU}}}{|\mathbf{A}||\mathbf{B}_{\mathrm{EMU}}|} = \frac{\lambda_1 b_1}{\sqrt{\sum_{i=1}^{N}\lambda_i^2(\bar{b} + \Delta b_i)^2}}$$
$$= \frac{\lambda_1 b_1}{\bar{b}\sqrt{n_r}} \left[1 + \frac{1}{n_r}\sum_{i=1}^{N}\lambda_i^2\left[\frac{2\Delta b_i}{\bar{b}} + \left(\frac{\Delta b_i}{\bar{b}}\right)^2\right]\right]^{-1/2}. \tag{16}$$

In the limit of $\Delta b \ll \bar{b}$, noting that $\lambda_i^2 = \lambda_i$,

$$\frac{\cos\theta_{\mathrm{EMU}}}{\cos\theta_0} = \lambda_1\sqrt{\frac{N}{n_r}} \left[1 - \frac{1}{n_r}\sum_{i=1}^{N}\lambda_i\epsilon_i\right] + O(\epsilon_i^2)$$
$$= \lambda_1\sqrt{\frac{N}{n_r}} \left[1 - \frac{\zeta}{n_r}\right] + O(\epsilon_i^2). \tag{17}$$

Here, $\epsilon_i \equiv \Delta b_i/\bar{b}$ and $\zeta \equiv \sum_{i=1}^{N}\lambda_i\epsilon_i$, and we derived the first equation by the Taylor expansion up to the first-order in $\epsilon_i$. According to the central limit theorem, $\zeta$ follows the Gaussian distribution in the case of sufficiently large $N$, and hence $\cos\theta_{\mathrm{EMU}}$ adheres to Gaussian distribution centered around $\sqrt{N/n_r}\cos\theta_0$, as long as $\lambda_1 \neq 0$. $\qquad\square$

## D  Proof of Lemma 3.4

*Proof.* First, we consider the case of the real-world data case with only limited data. According to Theorem 3.3 and Equation 6, the negative sample distribution can be written as a linear summation of an isotropic distribution as:

$$\tilde{p}_{n,\mathrm{EMU}}(\tilde{\mathbf{z}}_{\mathrm{EMU}}|v^{hr}) = \sum_{i=1}^{N} f_{\mathrm{iso}}(\Delta\mathbf{z}^\lambda|\mathbf{z}^{t,+}),$$
$$\mathbf{z}^{t,+} \sim \tilde{p}_d(\mathbf{z}^{t,+}|\mathbf{v}^{hr}), \tag{18}$$

where $\tilde{p}_n, \tilde{p}_d$ are the expression of $p_n$ and $p_d$ in the data space and $f_{\mathrm{iso}}(x|y)$ is an isotropic function around $y$. Note that in the above equation, it is assumed the following relation is always satisfied: $\tilde{z}_{\mathrm{EMU}} = \mathbf{z}^{t,+} + \Delta\mathbf{z}^\lambda$. If moving back to the idealized data space, Equation 18 reduces to:

$$p_{n,\mathrm{EMU}}(\tilde{\mathbf{z}}_{\mathrm{EMU}}|v^{hr})$$
$$= A \int d\mathbf{z}^{t,+} f_{\mathrm{iso}}(\Delta\mathbf{z}^\lambda|\mathbf{z}^{t,+})p_d(\mathbf{z}^{t,+}|\mathbf{v}^{hr}), \tag{19}$$

where $A$ is a constant. Then, we obtain the Equation 3.4 if explicitly demanding the relation: $\tilde{z}^{\mathrm{EMU}} = \mathbf{z}^{t,+} + \Delta\mathbf{z}^\lambda$, by considering the Dirac delta function. Note that the Equation 3.4 also assume that $f_{\mathrm{iso}}$ is a rapidly decaying function with a typical scale $\Delta\bar{z}$ which is the scale implicitly given in Proposition B.1. $\quad\square$

# E Unbounded Label Smoothing for Cross-Entropy Loss Function

Label smoothing is a well-known technique used to regularize classifier models (Szegedy et al., 2016). It is originally proposed to address the overconfidence issue that certain classifiers such neural networks may exhibit during training. It works by smoothing the class label as follows:

$$\mathbf{y}_{\text{LS}} = \hat{\mathbf{y}}(1 - \beta_{\text{LS}}) + \beta_{\text{LS}}/K, \tag{20}$$

where $\hat{\mathbf{y}} = \{y_0, y_1, ..., y_K\}$ is the *one-hot* label encoding, $\beta_{\text{LS}}$ is a label smoothing parameter that controls the model confidence, and $K$ is the number of classes. Note that the resultant smoothed label maintains the total sum equal to unity. However, when applied to problems with a high number of classes, label smoothing leads to small values for the negative class label (or elements for the contrastive learning case), which still induces an overly strong penalty on the EMU negative samples whose vector component include the true positive sample vector that should not be penalized. To address this issue, we propose a new approach called *Unbounded Label Smoothing* (Unbounded-LS), which is defined as follows:

$$y_{\text{ULS},k} = \begin{cases} 1 & \text{if} \quad k \in (+), \\ \beta & \text{otherwise}, \end{cases} \tag{21}$$

where $\beta$ is the softening parameter over the negative samples. The above modification of the negative sample labels does not affect the probabilistic interpretation of the model output, as it does not change the model output itself. Our unbounded LS discourages the model from penalizing the negative samples excessively.

# F Datasets

| Dataset | #entities | #relations | #triples |
|---------|-----------|------------|----------|
| FB15K-237 | 14,541 | 237 | 310,079 |
| YAGO3-10 | 123,188 | 37 | 1,179,040 |
| WN18-RR | 40,943 | 11 | 93,003 |

Table 4: Knowledge Graph dataset statistics. *training*, *validation* and *testing* refer to the number of triples under each split.

**FB15k-237** (Toutanova et al., 2015) is a commonly used benchmark for Knowledge Graph link prediction tasks and a subset of Freebase Knowledge Base (Bollacker et al., 2008). FB15k-237 was created as a replacement for FB15k, a previous benchmark that was widely adapted until the dataset's quality came into question in subsequent work (Toutanova et al., 2015) due to an excess of inverse relations.

**YAGO3-10** (Mahdisoltani et al., 2013) is a subset of YAGO (Yet Another Great Ontology)(Suchanek et al., 2007), a large semantic knowledge base that augments WORDNET and which was derived from WIKIPEDIA (Wikipedia contributors, 2004), WORDNET (Miller, 1995), WIKIDATA (Ahmadi & Papotti, 2021), and other sources. Because of its origins, YAGO entities are linked to WIKIDATA and WORDNET entity types. The dataset contains information about individuals, such as citizenship, gender, profession, as well as other entities such as organizations and cities. The subset YAGO3-10 contains triples with entities that have more than 10 relations.

**WN18RR** (Dettmers et al., 2018) is a link prediction dataset created from WN18 (Bordes et al., 2013), which is a subset of WORDNET, a popular large lexical database of English nouns, verbs, adjectives and adverbs. WORDNET contains information about relations between words, such as `hyponyms`, `hypernyms` and `synonyms` (Miller, 1995). However, similarly to the issues that occurred in FB15K, many test triples in WN18 are obtained by inverting triples from the training set. Therefore, WN18RR dataset was created in the same work as FB15K-237, in order to make a more challenging benchmark for link prediction.

## G    Experiment Setup

**Training Settings**   Here we describe the general settings we used to train all the models. The optimization was performed using Adam (Kingma & Ba, 2014) for $10^5$ iterations[9] with 256 negative samples[10]. The hyper-parameter tuning was performed with Optuna (Akiba et al., 2019). During the training, we monitored the loss over the validation set and selected the best model based on its performance on the validation set. For models trained with SAN negative samples, we utilized the default training setup from (Ahrabian et al., 2020).

**Evaluation Settings**   To ensure that all the methods were evaluated under the same conditions, we utilized standard metrics to report results, specifically the Mean Reciprocal Rank (MRR) and Hits at K (H@K). If multiple true tails exist for the same (head, relation)-pair, we filtered out the other true triplets at test time. To minimize model uncertainty resulting from random seeds or multi-threading, we performed three trials for each experiment and reported the mean and standard deviation of the evaluation scores.

**Baselines**   Among all existing baselines, we consider vanilla uniform negative sampling (Bordes et al., 2013) and SAN (Ahrabian et al., 2020) as the most relevant to compare our work against. Additionally, we included NSCaching (Zhang et al., 2019) as another baseline method, with its results provided in Appendix I [11]. The hyper-parameters for the baselines and EMU are tuned by utilizing Optuna (Akiba et al., 2019).

**Implementation Settings**   We modified the code originally developed by Ahrabian et al. (2020) to perform MIXUP and EMU with SAN. As explained in section 3, the models are trained using cross-entropy losses, incorporating one true tail sample and multiple negative samples. The optimization was performed using Adam (Kingma & Ba, 2014). The L3-norm loss function is used on the embedding vectors for the models with the vanilla uniform negative sampling and SAN. The mini-batch size is set to 1000. To compute the embedded triplets for all the KG models, we used an Embedding layer with a hidden dimension of : $d = 100$. A more detailed hyper-parameters are provided in Table 5 and Table 6. We tuned our hyperparameters, including the learning rate and the coefficient for weight-decay for baseline scores, through 10000 iterations on the FB15K-237 validation dataset using Optuna (Akiba et al., 2019). We adopted the officially provided configuration for the HAKE model setup (Zhang et al., 2020). The hyperparameters for EMU are detailed in Table 7.

**Computational Resources**   All the experiments other than HAKE were performed on one Nvidia GeForce GTX 1080 Ti GPU for each run. The experiments with HAKE were performed on one Nvidia GeForce RTX 3090 GPU for each run. The models were implemented by PyTorch 2.1.0 with CUDA11.8. Because of the additional gradient flow through negative sample components due to mutation, the total GPU memory usage of EMU becomes around 1.5 to 2 times larger than vanilla case in the case of DistMult with $(d, n) = (100, 256)$ where $d, n$ denote embedding dimension and negative sample number. Concerning the training time, although one-step duration becomes around twice longer[12], we also found that EMU reaches its maximum performance much earlier than the case without EMU, leading to a shorter training time in the end. We emphasize that the additional computational resource allows us to achieve a comparable performance with the case using even 10 times larger embedding dimension, which requires at least 10 times larger computational resource (see also subsection 4.3). Note that EMU does not require additional computational resources during inference.

---

[9]The total iteration number is the same as the one used in the SAN repository (Ahrabian et al., 2020) to reproduce their best result.

[10]In Appendix J, the influence of the number of negative samples on the outcomes is analyzed, and it is demonstrated that EMU outperforms the uniform-sampling approach in almost all instances.

[11]Due to the inherent intricacy involved in assessing the impact of different implementations (specifically, SAN-based and NSCaching-based codes) on performances, we relocated he results obtained with NSCaching to the Appendix.

[12]The length tends to increase with model complexity and size. However, Table 1 indicates that the performance gain by EMU also increases with the model complexity, as shown with HAKE

| Model | Method | Learning Rate | $\alpha$ | $n_{\mathrm{P}}/d$ | $\beta$ | $\gamma$ |
|---|---|---|---|---|---|---|
| ComplEX | uni | 0.1 | n/a | n/a | n/a | $10^{-5}$ |
| | SAN | 0.1 | n/a | n/a | n/a | $10^{-5}$ |
| | SAN_EMU | 0.1 | 0.34 | 0.92 | 0.12 | 0 |
| | uni_EMU | 0.1 | 0.34 | 0.92 | 0.12 | 0 |
| DistMult | uni | 0.1 | n/a | n/a | n/a | $10^{-5}$ |
| | SAN | 0.1 | n/a | n/a | n/a | $10^{-5}$ |
| | SAN_EMU | 0.1 | 0.73 | 0.94 | 0.25 | 0 |
| | uni_EMU | 0.1 | 0.73 | 0.94 | 0.25 | 0 |
| RotatE | uni | 0.005 | n/a | n/a | n/a | $10^{-3}$ |
| | SAN | 0.005 | n/a | n/a | n/a | $10^{-3}$ |
| | SAN_EMU | 0.005 | 0.11 | 0.39 | 0.53 | 0 |
| | uni_EMU | 0.005 | 0.11 | 0.39 | 0.53 | 0 |
| TransE | uni | 0.005 | n/a | n/a | n/a | $10^{-3}$ |
| | SAN | 0.005 | n/a | n/a | n/a | $10^{-3}$ |
| | SAN_EMU | 0.005 | 0.11 | 0.39 | 0.53 | 0 |
| | uni_EMU | 0.005 | 0.11 | 0.39 | 0.53 | 0 |

Table 5: Hyper-Parameters for FB15K-237 and WN18RR dataset. $\alpha, \beta, \gamma$ are the coefficient of original Loss, negative label value of Unbounded LS, and the coefficient of L3-norm loss, respectively.

| Model | Method | Learning Rate | $\alpha$ | $n_{\mathrm{P}}/d$ | $\beta$ | $\gamma$ |
|---|---|---|---|---|---|---|
| ComplEX | uni | 0.1 | n/a | n/a | n/a | $10^{-5}$ |
| | SAN | 0.1 | n/a | n/a | n/a | $10^{-5}$ |
| | SAN_EMU | 0.1 | 0.536 | 0.804 | 0.193 | 0 |
| | uni_EMU | 0.1 | 0.536 | 0.804 | 0.193 | 0 |
| DistMult | uni | 0.1 | n/a | n/a | n/a | $10^{-5}$ |
| | SAN | 0.1 | n/a | n/a | n/a | $10^{-5}$ |
| | SAN_EMU | 0.1 | 0.54 | 0.949 | 0.22 | 0 |
| | uni_EMU | 0.1 | 0.54 | 0.949 | 0.22 | 0 |
| RotatE | uni | 0.1 | n/a | n/a | n/a | $10^{-5}$ |
| | SAN | 0.1 | n/a | n/a | n/a | $10^{-5}$ |
| | SAN_EMU | 0.1 | 0.46 | 0.73 | 0.84 | 0 |
| | uni_EMU | 0.1 | 0.46 | 0.73 | 0.84 | 0 |
| TransE | uni | 0.1 | n/a | n/a | n/a | $5 \times 10^{-5}$ |
| | SAN | 0.1 | n/a | n/a | n/a | $5 \times 10^{-5}$ |
| | SAN_EMU | 0.1 | 0.11 | 0.39 | 0.53 | 0 |
| | uni_EMU | 0.1 | 0.11 | 0.39 | 0.53 | 0 |

Table 6: Hyper-Parameters for YAGO3 dataset. $\alpha, \beta, \gamma$ are the coefficient of original Loss, negative label value of Unbounded LS, and the coefficient of L3-norm loss, respectively.

# H  A Full Description of Main Result

In Table 8 we provide the full description of our result visualized in Figure 2. For the HAKE model, we provide the quantitative results in Table 9.

| Dataset | Method | $d$ | $n_s$ | $n_b$ | Max-Step | Learning Rate | $\alpha$ | $n_{\mathrm{P}}/d$ | $\beta$ |
|---|---|---|---|---|---|---|---|---|---|
| FB15K-237 | uni | 1000 | 256 | 1024 | 100000 | $5 \times 10^{-5}$ | n/a | n/a | n/a |
| | uni_EMU | | | | | $5 \times 10^{-5}$ | 0.1 | 0.128 | 0.964 |
| WN18RR | uni | 500 | 1024 | 512 | 80000 | $5 \times 10^{-5}$ | n/a | n/a | n/a |
| | uni_EMU | | | | | $5 \times 10^{-5}$ | 0.1 | 0.128 | 0.964 |
| YAGO3-10 | uni | 500 | 256 | 1024 | 180000 | $2 \times 10^{-4}$ | n/a | n/a | n/a |
| | uni_EMU | | | | | $2 \times 10^{-4}$ | 0.1 | 0.128 | 0.964 |

Table 7: Hyper-Parameters of HAKE model. $d, n_s, n_b$ denote hidden dimension size, negative sample number, and mini batch size, respectively. $\alpha, \beta, \gamma$ are the coefficient of original Loss, negative label value of Unbounded LS, and the coefficient of L3-norm loss, respectively.

| Model | Method | FB15K-237 MRR | HITS@10 | WN18RR MRR | HITS@10 | YAGO3-10 MRR | HITS@10 |
|---|---|---|---|---|---|---|---|
| ComplEX | uni | $0.306^{\pm 0.001}$ | $0.486^{\pm 0.000}$ | $0.461^{\pm 0.000}$ | $0.526^{\pm 0.002}$ | $0.399^{\pm 0.004}$ | $0.605^{\pm 0.003}$ |
| (Lacroix et al., 2018) | SAN | $0.275^{\pm 0.000}$ | $0.437^{\pm 0.001}$ | $0.467^{\pm 0.001}$ | $0.530^{\pm 0.001}$ | $0.318^{\pm 0.002}$ | $0.496^{\pm 0.004}$ |
| | SAN_EMU | $0.298^{\pm 0.001}$ | $0.474^{\pm 0.001}$ | $0.466^{\pm 0.002}$ | $0.543^{\pm 0.003}$ | $0.385^{\pm 0.002}$ | $0.563^{\pm 0.002}$ |
| | uni_EMU | $\mathbf{0.344}^{\pm 0.001}$ | $\mathbf{0.532}^{\pm 0.001}$ | $\mathbf{0.473}^{\pm 0.003}$ | $\mathbf{0.547}^{\pm 0.002}$ | $\mathbf{0.437}^{\pm 0.001}$ | $\mathbf{0.638}^{\pm 0.004}$ |
| DistMult | uni | $0.299^{\pm 0.001}$ | $0.476^{\pm 0.001}$ | $0.428^{\pm 0.001}$ | $0.489^{\pm 0.002}$ | $0.345^{\pm 0.001}$ | $0.538^{\pm 0.004}$ |
| (Yang et al., 2015) | SAN | $0.259^{\pm 0.001}$ | $0.415^{\pm 0.001}$ | $0.425^{\pm 0.001}$ | $0.481^{\pm 0.002}$ | $0.251^{\pm 0.002}$ | $0.428^{\pm 0.001}$ |
| | SAN_EMU | $0.282^{\pm 0.001}$ | $0.446^{\pm 0.002}$ | $0.427^{\pm 0.001}$ | $0.506^{\pm 0.004}$ | $0.293^{\pm 0.002}$ | $0.478^{\pm 0.002}$ |
| | uni_EMU | $\mathbf{0.332}^{\pm 0.001}$ | $\mathbf{0.513}^{\pm 0.001}$ | $\mathbf{0.446}^{\pm 0.002}$ | $\mathbf{0.523}^{\pm 0.003}$ | $\mathbf{0.403}^{\pm 0.004}$ | $\mathbf{0.601}^{\pm 0.004}$ |
| RotatE | uni | $0.305^{\pm 0.001}$ | $0.484^{\pm 0.001}$ | $\mathbf{0.458}^{\pm 0.001}$ | $\mathbf{0.549}^{\pm 0.002}$ | $0.378^{\pm 0.003}$ | $0.569^{\pm 0.003}$ |
| (Sun et al., 2019) | SAN | $0.257^{\pm 0.001}$ | $0.418^{\pm 0.001}$ | $0.456^{\pm 0.001}$ | $0.532^{\pm 0.003}$ | $0.303^{\pm 0.003}$ | $0.459^{\pm 0.003}$ |
| | SAN_EMU | $0.282^{\pm 0.000}$ | $0.455^{\pm 0.001}$ | $0.451^{\pm 0.001}$ | $0.516^{\pm 0.002}$ | $0.363^{\pm 0.002}$ | $0.535^{\pm 0.002}$ |
| | uni_EMU | $\mathbf{0.329}^{\pm 0.001}$ | $\mathbf{0.514}^{\pm 0.001}$ | $0.453^{\pm 0.002}$ | $0.525^{\pm 0.002}$ | $\mathbf{0.391}^{\pm 0.001}$ | $\mathbf{0.609}^{\pm 0.002}$ |
| TransE | uni | $0.314^{\pm 0.001}$ | $0.479^{\pm 0.002}$ | $0.227^{\pm 0.002}$ | $0.506^{\pm 0.002}$ | $0.233^{\pm 0.001}$ | $0.389^{\pm 0.005}$ |
| (Bordes et al., 2013) | SAN | $0.299^{\pm 0.001}$ | $0.460^{\pm 0.002}$ | $\mathbf{0.237}^{\pm 0.001}$ | $\mathbf{0.518}^{\pm 0.002}$ | $0.222^{\pm 0.002}$ | $0.375^{\pm 0.001}$ |
| | SAN_EMU | $0.281^{\pm 0.000}$ | $0.450^{\pm 0.003}$ | $0.202^{\pm 0.001}$ | $0.493^{\pm 0.001}$ | $0.221^{\pm 0.003}$ | $0.383^{\pm 0.001}$ |
| | uni_EMU | $\mathbf{0.323}^{\pm 0.001}$ | $\mathbf{0.503}^{\pm 0.003}$ | $0.216^{\pm 0.001}$ | $0.493^{\pm 0.001}$ | $\mathbf{0.255}^{\pm 0.002}$ | $\mathbf{0.438}^{\pm 0.002}$ |

Table 8: MRR and Hit@10 of the results on FB15K-237, WN18RR, and YAGO3-10 datasets. "uni" means the uniform negative sampling, "SAN" means the structure aware negative sampling. The shown results are the average with the standard deviation of three trials of the randomly determined initial weights.

| Model | Method | FB15K-237 MRR | HITS@10 | WN18RR MRR | HITS@10 | YAGO3-10 MRR | HITS@10 |
|---|---|---|---|---|---|---|---|
| HAKE | uni | $0.308^{\pm 0.002}$ | $0.493^{\pm 0.000}$ | $0.436^{\pm 0.002}$ | $0.487^{\pm 0.003}$ | $0.452^{\pm 0.005}$ | $0.651^{\pm 0.004}$ |
| (Zhang et al., 2020) | uni_EMU | $\mathbf{0.316}^{\pm 0.001}$ | $\mathbf{0.503}^{\pm 0.001}$ | $\mathbf{0.453}^{\pm 0.001}$ | $\mathbf{0.526}^{\pm 0.002}$ | $\mathbf{0.499}^{\pm 0.000}$ | $\mathbf{0.687}^{\pm 0.001}$ |

Table 9: MRR and Hit@10 of the results of HAKE model trained on FB15K-237, WN18RR, and YAGO3-10 datasets. "uni" means the uniform negative sampling. The shown results are the average with the standard deviation of three trials of the randomly determined initial weights.

# I   Results using NScaching

This section presents the results obtained with EMU and NSCaching (Zhang et al., 2019)[13]. We modified the official NSCaching repository to enable the use of the cross entropy loss function and EMU. We used the same hyperparameters as those provided in section 3, mini-batch size is 1000 and 256 negative samples.

---

[13]We provided the results with NSCaching in the appendix rather than the main body because of differences in the implementation between the official repositories for SAN and NSCaching, which makes it difficult to compare those results equally.

| Model | Method | MRR | Hit@10 |
|-------|--------|-----|--------|
| ComplEX | NSCaching | $0.387^{\pm 0.001}$ | $0.577^{\pm 0.001}$ |
| | NSCaching_EMU | $\mathbf{0.394}^{\pm 0.001}$ | $\mathbf{0.585}^{\pm 0.005}$ |
| DistMult | NSCaching | $0.370^{\pm 0.001}$ | $0.557^{\pm 0.003}$ |
| | NSCaching_EMU | $\mathbf{0.376}^{\pm 0.002}$ | $\mathbf{0.565}^{\pm 0.000}$ |
| TransE | NSCaching | $0.322^{\pm 0.001}$ | $\mathbf{0.470}^{\pm 0.002}$ |
| | NSCaching_EMU | $\mathbf{0.323}^{\pm 0.001}$ | $0.467^{\pm 0.004}$ |
| RotatE | NSCaching | n/a | n/a |
| | NSCaching_EMU | n/a | n/a |

Table 10: MRR and Hit@10 of the results with NSCaching code trained using FB15K-237. "NSCaching" means the NSCaching negative smapling. The shown results are the average with the standard deviation of three trials of the randomly determined initial weights. Note that the result of RotatE is omitted because RotatE is not provided in the original repository.

EMU parameters are provided in Table 11. The results are provided in Table 10 which demonstrate that our EMU consistently improves the performance, even when using NSCaching[14].

| Model | Method | Learning Rate | $\alpha$ | $n_\mathrm{P}/d$ | $\beta$ | $\gamma$ |
|-------|--------|---------------|----------|------------------|---------|----------|
| ComplEX | uni | $3 \times 10^{-4}$ | n/a | n/a | n/a | $10^{-5}$ |
| | NSCaching | $3 \times 10^{-4}$ | n/a | n/a | n/a | $10^{-5}$ |
| | NSCaching_EMU | $3 \times 10^{-4}$ | 0.44 | 0.34 | 0.32 | 0 |
| DistMult | uni | $10^{-3}$ | n/a | n/a | n/a | $10^{-5}$ |
| | NSCaching | $10^{-3}$ | n/a | n/a | n/a | $10^{-5}$ |
| | NSCaching_EMU | $10^{-3}$ | 0.68 | 0.17 | 0.16 | 0 |
| TransE | uni | $5 \times 10^{-4}$ | n/a | n/a | n/a | $2 \times 10^{-2}$ |
| | NSCaching | $5 \times 10^{-4}$ | n/a | n/a | n/a | $2 \times 10^{-2}$ |
| | NSCaching_EMU | $5 \times 10^{-4}$ | 0.54 | 0.168 | 0.151 | 0 |

Table 11: Hyper-Parameters of NScaching code trained using FB15K-237 dataset. $\alpha, \beta, \gamma$ are the coefficient of EMU Loss, negative label value of Unbounded LS, and the coefficient of L3-norm loss, respectively.

## J   Negative Sample Number Dependence

In the main body of this work, we maintained a fixed number of negative samples at 256. However, in Figure 5, we depict the relationship between the optimal MRR and the number of negative samples employed. Our experiments were conducted using the FB15K-237. Notably, EMU demonstrated superior MRR values in most cases, with a notable increase in performance gains as the number of negative samples increased.

## K   Hyper-Parameter Dependence Study

In Table 12 illustrates the dependence of EMU performance on hyper-parameters: $\alpha, n_P/d$, and $\beta$. We considered the DistMult as a KGE model. The results indicate that the excessively large values of the coefficient of EMU loss, $\alpha$, are undesirable. Conversely, it is preferable to use a moderate value for the negative label value of Unbounded LS, $\beta$. Finally, the performance is relatively insensitive to the change of the mutation ratio, $n_P/d$, but exhibits a slight improvement as the value approaches the optimal one.

---

[14]The obtained MMR and H@10 values may appear excessively good; however, we believe that this may be partly due to the NSCaching code implementation, although we cannot confirm this with certainty.

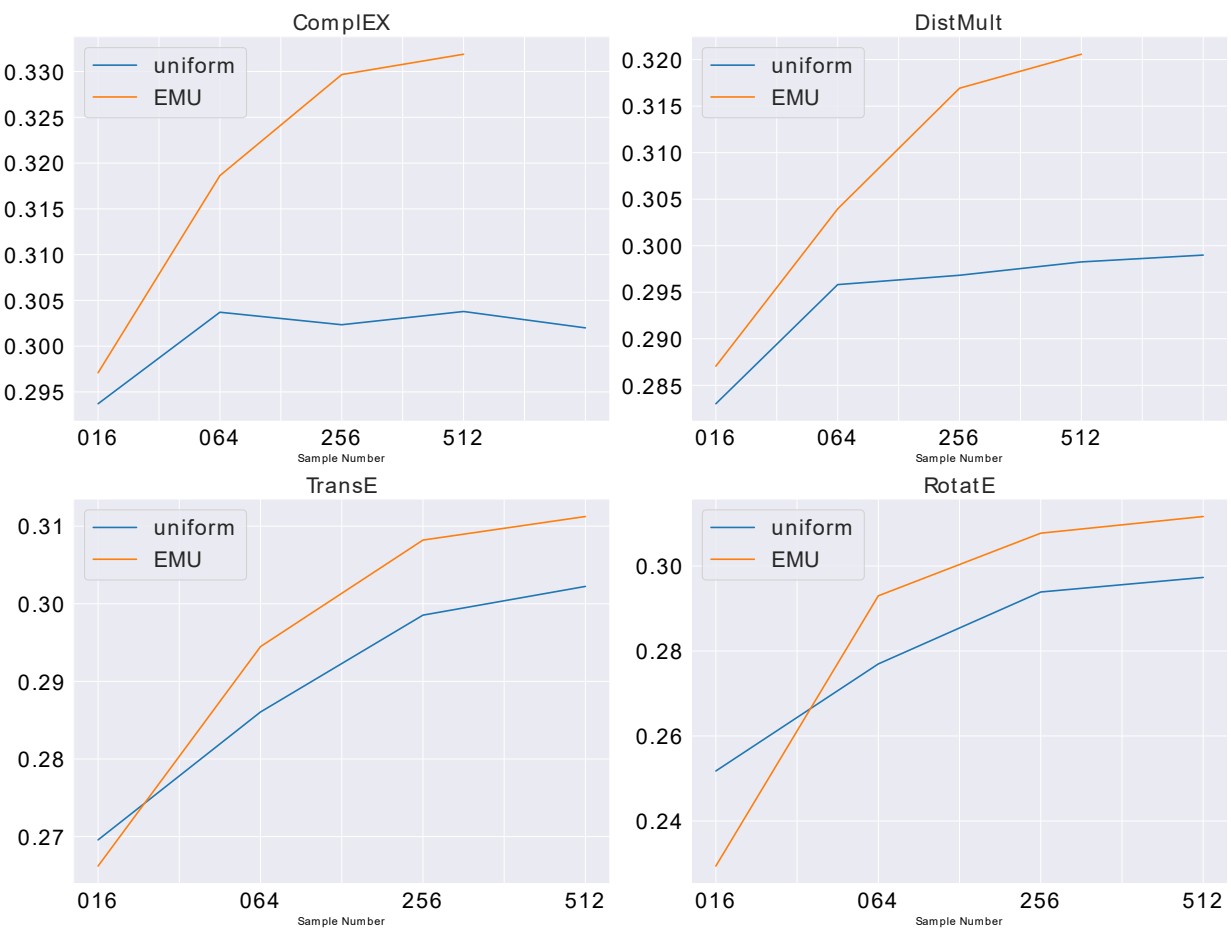

Figure 5: The negative sample number dependence of MRR trained on FB15K-237. The right-edge of the ComplEX and DistMult of the uniform negative sampling case is the "1 VS ALL" results.

In line with the above trend, for practical applications, we recommend initially testing the hyperparameters identified in this study, as outlined in Table 5, Table 6, and Table 11. In particular, attention should be given to tuning the coefficient of the EMU loss, $\alpha$. If computational resources permit, the use of hyperparameter optimization tools, such as Optuna (Akiba et al., 2019) and KGTuner (Zhang et al., 2022), is also advised.

| $(\alpha, n_{\mathrm{P}}/d, \beta)$ | MRR | HITS@10 |
|---|---|---|
| $(0.11, 0.914, 0.53)$  **baseline** | **0.333** | **0.513** |
| $(0.5, 0.914, 0.53)$ | 0.318 | 0.498 |
| $(0.9, 0.914, 0.53)$ | 0.306 | 0.484 |
| $(0.11, 0.1, 0.53)$ | 0.326 | 0.509 |
| $(0.11, 0.5, 0.53)$ | 0.327 | 0.504 |
| $(0.11, 0.914, 0.1)$ | 0.314 | 0.496 |
| $(0.11, 0.914, 0.9)$ | 0.326 | 0.501 |

Table 12: Hyper-Parameters study results using FB15K-237 dataset with DistMult model. $\alpha, \beta, n_P$ are the coefficient of EMU Loss, negative label value of Unbounded LS, and the number of mutation components, respectively.

| Model | Ablation | MRR | | HITS@10 | |
|-------|----------|-----|--|---------|--|
| ComplEX | EMU | **0.344** | | **0.532** | |
| | w/t ULS | 0.252 | (**-0.092**) | 0.411 | (**-0.121**) |
| | w/t Emb.Mut. | 0.302 | (-0.042) | 0.477 | (-0.055) |
| | w/t EMU | 0.306 | (-0.038) | 0.486 | (-0.046) |
| DistMult | EMU | **0.332** | | **0.513** | |
| | w/t ULS | 0.254 | (**-0.076**) | 0.415 | (**-0.098**) |
| | w/t Emb.Mut. | 0.300 | (-0.032) | 0.477 | (-0.036) |
| | w/t EMU. | 0.311 | (-0.021) | 0.489 | (-0.024) |
| RotatE | EMU | **0.329** | | **0.514** | |
| | w/t ULS | 0.236 | (**-0.093**) | 0.386 | (**-0.128**) |
| | w/t Emb.Mut. | 0.312 | (-0.017) | 0.496 | (-0.018) |
| | w/t EMU | 0.305 | (-0.024) | 0.484 | (-0.030) |
| TransE | EMU | **0.323** | | **0.503** | |
| | w/t ULS | 0.260 | (**-0.063**) | 0.423 | (**-0.080**) |
| | w/t Emb.Mut. | 0.308 | (-0.015) | 0.491 | (-0.012) |
| | w/t EMU | 0.314 | (-0.009) | 0.479 | (-0.024) |

Table 13: Ablation study results on FB15K-237. The number in the parentheses are the difference from the "EMU" results. "ULS" means Unbounded Label Smoothing, and "Emb.Mut." means Embedding Mutation.

## L  Ablation Study

In this subsection, we present the results of our ablation study to understand the individual contributions of the two main components of EMU, i.e. the Embedding Mutation and the Unbounded-LS. To achieve this goal, we used the FB15k-237 dataset as a reference benchmark and performed a set of experiments by decoupling the embedding mutation from the Unbounded-LS. We trained the KGE models under three different scenarios: 1) the *EMU*, which represents the proposed EMU combining Embedding Mutation and Unbounded-LS; 2) the baseline without Unbounded-LS (*w/t Unbounded-LS*); 3) the baseline without the Embedding Mutation (*w/t Emb.Mut.)*, and 4) the case without EMU (*w/t EMU*). The aim of the experiments was to compare the performance reduction by removing one of the components, thereby gaining insights into their relative importance.

The results obtained from the ablation study are presented in Table 13, with the first two columns indicating the target model and the experiment setup, and the last two columns showing the MRR and HITS@10 results with the performance loss compared to the baseline. Our results consistently demonstrate that Unbounded-LS has a strong impact on all models[15]. This is quite natural because EMU without Unbounded-LS penalizes not only pure negative samples but also true sample embedding because of Embedding Mutation. We also hypothesize that the effectiveness of Unbounded-LS can also stem from its ability to effectively allow large gradient flow values from negative samples. This is attributed to the relatively large negative sample labels (typically larger than 0.1) and the tendency of Embedding Mutation to create harder negatives, which results in larger loss values. The resulting gradients affect both the positive and negative sample components, ultimately leading to an improved representation of their embedding. In conclusion, Embedding Mutation combined with Unbounded-LS consistently (EMU) improves performance of multiple and diverse models.

---

[15]We also compared Unbounded-LS and vanilla LS (Szegedy et al., 2016) in Appendix M and found that Unbounded-LS is more effective than the usual LS

## M    Comparison between Vanilla LS and Unbounded LS

In this study we proposed the unbounded label-smoothing (LS) technique. To assess its efficacy, we also trained our models using vanilla LS (Szegedy et al., 2016) with a label smoothing parameter of 0.2. The result is provided in Table 14 which demonstrate the clear speriority of Unbounded LS for all cases.

| Model | Ablation | MRR | HITS@10 |
|---|---|---|---|
| ComplEX | Unbounded LS | **0.344** | **0.532** |
| (Trouillon et al., 2016) | Vanilla LS (w/t ULS) | 0.262  (**-0.082**) | 0.423  (**-0.109**) |
| DistMult | Unlabeled LS | **0.332** | **0.513** |
| (Yang et al., 2015) | Vanilla LS (w/t ULS) | 0.252  (**-0.080**) | 0.410  (**-0.103**) |
| RotatE | Unbounded LS | **0.329** | **0.514** |
| (Sun et al., 2019) | Vanilla LS (w/t ULS) | 0.236  (**-0.093**) | 0.382  (**-0.132**) |
| TransE | Unbounded LS | **0.322** | **0.503** |
| (Bordes et al., 2013) | Vanilla LS (w/t ULS) | 0.259  (**-0.063**) | 0.423  (**-0.080**) |

Table 14: A comparison between Unbounded LS and vanilla LS.

## N    An Brief Introduction to Mixup

In this section, we provide a brief introduction of *Mixup* (Zhang et al., 2018). MIXUP is a simple regularization technique that constructs virtual training examples as:

$$\tilde{\mathbf{z}}_{\text{Mixup}} \equiv \lambda \mathbf{z}_i + (1 - \lambda)\mathbf{z}_j, \tag{22}$$

where $\mathbf{z}_i, y_i$ are the $i$-th input and label data, $\lambda \sim \text{Beta}(\alpha, \alpha)$ is a random scalar value controlling mixing ratio between the two samples, and $\alpha \in (0, \infty)$. MIXUP is typically applied across the elements of a given batch, and randomly produces new virtual samples by linearly mixing two classes as shown in Equation 22. While MIXUP was originally proposed to address problems such as reducing memorization of corrupted labels and increasing the robustness to adversarial examples, we observed limitations to its performance when we extended it to embedding methods (refer to Table 2). We hypothesize that the linear nature of MIXUP-generated example restricts the magnitude of gradients without changing their direction, which limits its effectiveness. On the other hand, EMU overcomes this limitation by producing updates that can take multiple directions and thus, enhances model training.

For the MIXUP experiments, we simply replaced the embedding mutation into Mixup in Equation 22. For simplicity, we set $\lambda \sim \text{Beta}(\alpha, \beta)|_{\alpha=2, \beta=1}$. Note that here we did not set as $\alpha = \beta$, as in the original implementation (Zhang et al., 2018), because we found that using different values of $\alpha$ and $\beta$ resulted in a significantly improved accuracy. We attribute this to the skewed probabilistic distribution that arises due to the different values of $\alpha$ and $\beta$, which allows for a higher ratio of negative samples than positive samples in the mixed-tail embedding vectors.

## O    Detailed Explanation of the Application of EMU to NBFNet

| Model | $\alpha$ | $n_{\text{P}}/d$ | $\beta$ |
|---|---|---|---|
| NBFNet w.t. EMU | 1.0 | 0.94 | 0.25 |

Table 15: EMU hyper-parameter used for the experiments on NBFNet.

The experiments on Neural Bellman-Ford Networks (NBFNet) (Zhu et al., 2021), described in subsection 4.6, were conducted using the official repository[16]. We integrated the EMU loss function into the repository's

---

[16]https://github.com/DeepGraphLearning/NBFNet

code and executed the experiments using the default configuration. The hyperparameters for EMU are listed in Table 15. The experiments were conducted using three different random seeds, which were consistently applied to both the baseline and EMU experiments. All experiments were performed on a single NVIDIA A100 GPU. In this experiment, we used the binary cross entropy loss for EMU loss because of the usage of binary cross entropy loss for the vanilla NBFNet.

## P    Cosine Similarity Between Negative Samples

Figure 6 plots the cosine similarity among negative samples for EMU-EMU, uniform-uniform, and uniform-EMU. The results indicate that the similarity between uniform negative samples are consistently lower than that of EMU negative samples, suggesting that EMU generates more hard negative samples.

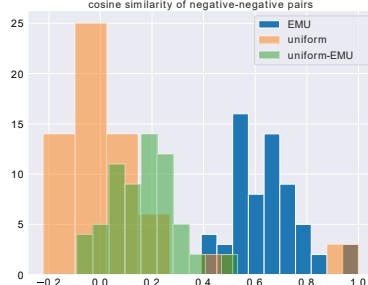

Figure 6: Cosine similarity of negative-negative tail pairs for DistMult with FB15k-237.

