# OpenReview forum: "Optimal Embedding Guided Negative Sample Generation for Knowledge Graph Link Prediction"
_TMLR — Accepted by TMLR_

### Review · Reviewer_VhgB · 2025-02-01

**Summary Of Contributions:**

This paper introduces Embedding Mutation (EMU), a framework for generating negative samples in knowledge graph embedding (KGE) link prediction. EMU mutates negative embeddings using components from positive embeddings, guided by theoretical conditions for optimal embedding. It is simple, compatible with existing methods, and consistently improves link prediction performance, matching models with larger embedding dimensions.

**Audience:**

Yes

**Claims And Evidence:**

Yes

**Requested Changes:**

1. The paper claims: "EMU is currently limited to knowledge graph link prediction models trained using the cross-entropy loss function." Can EMU be extended to other loss functions, such as margin-based loss functions, beyond cross-entropy?

2. Are there more effective and parameter-free methods to determine the optimal amount of embedding mutation rather than random sampling?

3. What further analysis can be done to understand why EMU performs differently on WN18rr compared to other datasets? Is there a link between the optimal distribution and the features of different knowledge graphs?

4. How can we evaluate potential bias that may arise during the model training process due to feature mutation?

5. The proposed EMU method offers a novel approach to generating high-quality negative samples, which may lead to integration with tools like KGTuner (arXiv:2205.02460) for enhanced knowledge graph learning. Considering that KGTuner is designed for efficient hyperparameter search in Knowledge Graph Learning, how can KGTuner and the proposed EMU method be effectively combined to achieve joint optimization?

6. Given that the proposed EMU method focuses on improving the quality of negative samples, how can EMU be effectively integrated with various existing KGE models (e.g., NBFNet (arXiv: 2106.06935), AdaProp (arXiv:2205.15319),  A*net (arXiv:2206.04798) and PPR (arXiv:2403.10231)).

**Strengths And Weaknesses:**

### Strengths

1. Novel Method. EMU introduces a novel and effective method for generating hard negative samples by mutating embedding vectors. This approach is different from conventional methods that focus on identifying challenging negatives within the training data.

2. Simplicity and Compatibility. EMU is designed to be simple and easy to integrate with existing KGE models and negative sampling strategies.

3. Significant Performance Improvement. The experimental results consistently show substantial performance improvements with EMU across various KGE models, datasets, and negative sampling methods.

4. Theoretical Foundation: The paper provides a theoretical analysis of the conditions for optimal negative sampling in KGE, which gives a solid basis for the proposed method, moving beyond purely empirical approaches.

5. Comprehensive Experimental Evaluation. The experiments are comprehensive, covering several datasets (FB15k-237, WN18RR, YAGO3-10) and multiple KGE models (ComplEX, DistMult, RotatE, TransE, HAKE), along with comparisons to several baselines.

### Weaknesses

1. Computational Overhead: The additional gradient flow through negative sample components due to mutation results in a larger GPU memory usage and training time, although the method tends to reach maximum performance faster than other methods.

2. Lack of generalizability: EMU did not perform well on WN18RR dataset, which requires further analysis.

---

> ### Author Response · Authors · 2025-02-19
> **Replies**
>
> Dear Reviewer VhgB,
>
> Thank you for your kind comments. The followings are our responses to your requests.
>
> $\textbf{W1 (Computational Cost)}$
>
> Thank you for pointing this out. While a slight increase in computational cost is inevitable, this is a common trade-off for methods incorporating auxiliary loss functions, which, as mentioned, enable performance improvements.
>
> $\textbf{R1 (Extension to Other Loss Functions)}$
>
> We appreciate your suggestion. As discussed in the General Reply, it is actually possible to use alternative loss functions. However, we did not conduct experiments with them because (binary) cross-entropy loss is widely recognized as the most effective choice for most models.
>
> $\textbf{R2 (Sampling Method for Mutation)}$
>
> As demonstrated in Theorem 3.3, random mutation results in an isotropic distribution of the generated negative samples. Since isotropic negative samples lead to optimal embedding, we believe that random sampling should be maintained as the mutation strategy.
>
> $\textbf{R3; W2 (Lack of Generalizability on WN18RR)}$
>
> We have already discussed the reason for EMU’s reduced effectiveness on WN18RR in Section 4.4 and Figure 4. In this dataset, negative samples are already isotropically distributed in the latent space, satisfying the optimal embedding condition even without EMU.
> Additionally, WN18RR contains broad semantic relations (e.g., synonyms, metonyms), leading to overlapping domains and ranges for negative samples. This overlap may reduce EMU’s effectiveness, as it limits the contrast between positive and negative samples.
>
> $\textbf{R4 (Potential Bias)}$
>
> Since EMU mutations are based on a bias-free random distribution, we do not expect EMU to introduce any additional bias to the model or data. Moreover, thanks to “generating” negative samples, we consider that EMU could help even mitigate false-negative penalization (i.e., missing true samples), a fundamental issue in traditional negative sampling methods.
>
> $\textbf{R5 (Potential Migration to KGTuner)}$
>
> Thank you for introducing this interesting hyperparameter tuning method. In principle, KGTuner could be integrated into our approach. However, given EMU’s simple structure, we opted for Optuna, a Bayesian optimization-based hyperparameter tuning method.
> A potential application of KGTuner could be:
>
> 1. Optimizing the base model’s hyperparameters using KGTuner.
> 2. Optimizing EMU-specific hyperparameters using KGTuner or another method.
>
> $\textbf{R6 (Application to Recent Models)}$
>
> Following the encouraging suggestion, we applied EMU to NBFNet, one of the SOTA models for knowledge graph embedding. The detailed results are provided in the General Reply, where EMU achieved possibly a new SOTA performance on FB15k-237.

---

> > ### Author Response · Authors · 2025-02-26
> >
> > Dear Reviewer VhgB,
> >
> > Thank you for your supportive comments. Please let us know if you have further discussion points to prepare your formal decision recommendation to the Action Editor.
> >
> > Best regards,
> >
> > Authors

---

> ### Comment · Action_Editor_Q6qJ · 2025-03-05
>
> Could you give your recommendation?

---

### Review · Reviewer_wpzR · 2025-02-01

**Summary Of Contributions:**

The paper presents a new method for improving knowledge graph embedding (KGE) models by generating high-quality negative samples. The method, called Embedding Mutation (EMU), generates negative samples by replacing a certain amount of the embedding vector components in the negative sample with the corresponding parts of the true positive vector components. The work implements the method based on DistMult and demonstrates that EMU can help DistMult achieve optimal embeddings. Experiments on benchmark datasets show that the proposed EMU method can be successfully integrated into several KGE methods to achieve performance improvements.

**Audience:**

Yes

**Broader Impact Concerns:**

NA.

**Claims And Evidence:**

Yes

**Requested Changes:**

- Discussions on whether the theory still holds when integrated with other KGE models or what the prerequisites are for the proposed theory to hold.

- It would be beneficial to include experiments with more recent KGE models, if possible.

**Strengths And Weaknesses:**

Strengths:

- I agree that negative samples are crucial for KGE training. The proposed method generates negative samples by mutating embeddings, which differs from conventional methods that corrupt positive triples. The simplicity of EMU allows for seamless integration with existing KGE models and negative sampling methods.

- The paper provides a solid theoretical basis for the proposed method, identifying conditions under which negative samples lead to optimal KGE for DistMult.

- The work conducts extensive experiments across multiple datasets and KGE models, demonstrating some improvements in link prediction performance.

Weaknesses:

- It seems that the theoretical analysis in the paper is based on DistMult. It remains to be seen whether the theory still holds when integrated with other KGE models.

- In my view, achieving optimal learning objectives does not imply that the vectors are optimal, as the non-existent edges in the KG may not necessarily be incorrect. The theoretical analysis and referenced work in this paper do not take this point into consideration.

Typos:

- In Abstract, "consistenly" -> "consistently".

---

> ### Author Response · Authors · 2025-02-19
> **Replies**
>
> Dear Reviewer wpzR,
>
> Thank you for your kind comments. The followings are our responses to your requests.
>
> $\textbf{W1; R1 (Extension of Theoretical Analysis to Other KGE Models)}$
>
> As discussed in the General Reply, we found that our theoretical analysis can be generalized to a more flexible formulation, $f(z^h, z^r, z^t)$, which encompasses most KGE models. Thank you very much for your encouraging feedback!
>
>
> $\textbf{W2 (Difference Between Optimal Embedding and Best Performance Models)}$
>
> While optimal embedding does not necessarily guarantee maximal prediction performance (as noted by Reviewer wpzR), EMU ensures optimal embedding with respect to the assumed loss function. Additionally, by generating negative samples isotropically around the target positive sample, EMU helps mitigate the issue of false negatives (i.e., missing true samples), which is a fundamental limitation of traditional negative sampling methods.
> This theoretical insight is further empirically validated in both our original experiments and the newly conducted experiments on NBFNet.
>
>
> $\textbf{R2 (Application to Recent Models)}$
>
> We have validated EMU’s effectiveness on NBFNet, as discussed in the General Reply, demonstrating possibly a new SOTA performance when applied to NBFNet. Please refer to the updated results for further details.

---

> > ### Comment · Reviewer_wpzR · 2025-02-20
> >
> > Thank you for addressing my concerns.
> >
> > I noticed that you added a new section titled "Application to NBFNet" to the paper. However, this section is quite short. Perhaps it would be better to merge it into the "Empirical Validation" section (minor suggestion).

---

> > > ### Author Response · Authors · 2025-02-20
> > > **reply**
> > >
> > > Thank you very much for your kind suggestion! Following your advice, we merged the section describing EMU + NBFNet into Section 4 (Empirical Validation), as Section 4.6.

---

> ### Comment · Action_Editor_Q6qJ · 2025-03-05
>
> Could you give your recommendation?

---

> > ### Comment · Reviewer_wpzR · 2025-03-06
> >
> > I have no additional concerns and recommend this paper for publication. Thanks.

---

### Review · Reviewer_2EYK · 2025-02-12

**Summary Of Contributions:**

This paper attempts to address the challenge of generating high-quality negative samples for training Knowledge Graph Embedding (KGE) models, which is crucial for improving link prediction performance. The authors theoretically investigate the conditions under which negative samples lead to optimal KGE and propose a novel framework called Embedding MUtation (EMU). EMU generates negative samples by mutating embedding vectors, ensuring that the generated samples satisfy the identified optimal condition. The paper demonstrates through extensive experiments that EMU significantly improves link prediction performance across various KGE models and datasets, even enabling models with smaller embedding dimensions to achieve performance comparable to those with much larger dimensions.

**Audience:**

Yes

**Claims And Evidence:**

Yes

**Requested Changes:**

See Weaknesses

**Strengths And Weaknesses:**

## Strengths:
1. The paper provides a solid theoretical analysis, identifying a sufficient condition for an effective negative sample distribution that leads to optimal KGE.
2. The proposed EMU framework generates negative samples by mutating embedding vectors, and is designed to be simple and compatible with existing KGE models, making it easy to integrate into current workflows. This practical aspect enhances its potential for widespread adoption.
3. The paper includes extensive experiments across multiple datasets and KGE models, demonstrating consistent and significant improvements in link prediction performance. The results show that EMU can achieve performance comparable to models with much larger embedding dimensions, reducing computational complexity.
4. The authors provide an implementation of EMU and the experiments, which facilitates reproducibility and further research by the community.

## Weaknesses:
1. The core of EMU is Eq.(6), which is pretty simple and straightforward. While the authors already provide a theoretical analysis on its relationship with Y20, I think their relationships are not as close as claimed in the paper, and it would be better if the authors could provide more insights on why this simple mutation strategy works well in practice. For example, what kind of information is preserved in the mutated embeddings that make them effective negative samples?
2. Some experiments are not convincing. For example, in Table 1, the ablation study was conducted with only one dataset and two baseline models, then the authors concluded that EMU enables models with smaller embedding dimensions to achieve performance comparable to those with much larger dimensions. However, the results may not be generalizable to other datasets or models. It would be better if the authors could conduct more ablation studies on different datasets and models to validate their conclusions. Besides, larger dimensions don't always result in better performance.
3. It's pleased to see that the authors have compared with Mixup in KGE tasks. Mixup is a general data augmentation method, and it's not specifically designed to generate negative samples in KGE tasks. Regardless of its theoretical foundation, EMU also appears to be general, but this work only discusses its application in KGE tasks. It would be better if the authors could provide more insights on how EMU could be applied to other tasks or domains.

---

> ### Author Response · Authors · 2025-02-19
> **Replies**
>
> Dear Reviewer 2EYK,
>
> Thank you for your kind comments. The followings are our responses to your requests.
>
> $\textbf{W1 (Theoretical Insights)}$
>
> >  I think their relationships are not as close as claimed in the paper, and it would be better if the authors could provide more insights on why this simple mutation strategy works well in practice.
>
> We appreciate the reviewer’s insightful comment; however, we respectfully disagree with this assessment. As stated in the paper, we have already provided a rigorous theoretical analysis supporting our claims. To clarify, we summarize the key points of our analysis as follows:
>
> * The isotropic distribution of negative samples around the target positive sample leads to an optimal embedding.
> * EMU generates negative samples that are isotropically distributed around the target positive sample.
> * While the optimal embedding does not necessarily guarantee maximal prediction performance (as pointed out by Reviewer wpzR), EMU ensures optimal embedding with respect to the assumed loss function, at least.
> * Moreover, thanks to the fact that EMU “generates” new negative samples, EMU could mitigate the issue of false negatives (i.e., missing true samples), which is a fundamental limitation of other negative sampling methods.
>
> These theoretical insights are further empirically validated in both our original experiments and the new experiments on NBFNet.
>
> $\textbf{W2 (Comprehensiveness of the Ablation Study):}$
>
> > It would be better if the authors could conduct more ablation studies on different datasets and models to validate their conclusions.
>
> We would like to highlight that extensive ablation studies have already been provided in Appendix L, covering multiple models. Regarding dataset diversity, we chose FB15k-237 as our benchmark dataset because it is widely studied in the research community, and most prior works—including NBFNet—typically conduct ablation studies on a single dataset.
> It is important to emphasize that the goal of the ablation study is to provide insights into the impact of our method, rather than to conduct an exhaustive case study across various datasets. We believe that including additional datasets may introduce unnecessary complexity and potentially dilute the clarity of our findings.
>
> > Besides, larger dimensions don't always result in better performance.
>
> While we acknowledge that increasing dimensionality does not always lead to better performance in general, our experiments consistently show that higher dimensions improve performance. Additionally, EMU significantly enhances the performance of lower-dimensional models, reinforcing its effectiveness.
>
> $\textbf{W3 (Potential Applications of EMU)}$
>
> We sincerely appreciate the reviewer’s encouraging suggestions. Due to EMU’s simple and flexible structure, it can theoretically be applied to other tasks, such as graph node classification. Additionally, it has the potential to serve as an alternative to latent vector Mixup in various applications. We have incorporated this discussion in the revised Discussion section of the paper.

---

> > ### Author Response · Authors · 2025-02-26
> >
> > Dear Reviewer 2EYK,
> >
> > Thank you for your supportive comments. Please let us know if you have further discussion points to prepare your formal decision recommendation to the Action Editor.
> >
> > Best regards,
> >
> > Authors

---

> > ### Comment · Reviewer_2EYK · 2025-02-27
> >
> > Dear authors,
> > Thank you for your responses, which partially address my concerns. However, I still have unresolved issues that require further clarification:
> > > As stated in the paper, we have already provided a rigorous theoretical analysis supporting our claims. To clarify, we summarize the key points of our analysis as follows:
> >
> > I appreciate the theoretical framework for EMU and "optimal embedding", but the **connections to Y20** are still unclear. Y20 is claimed to be the foundation of EMU, but Section 3.1 appears somewhat independent from the other sections. Hence, the existence of Section 3.1 (as well as the corresponding motivation) is arguable. Please provide more information on this.
> >
> > > We would like to highlight that extensive ablation studies have already been provided in Appendix L, covering multiple models.
> >
> > I'm aware of the experiments in Appendix L, but they are not complementary to the experiments in Table 1.
> >
> > > Regarding dataset diversity, we chose FB15k-237 as our benchmark dataset because it is widely studied in the research community, and most prior works—including NBFNet—typically conduct ablation studies on a single dataset. It is important to emphasize that the goal of the ablation study is to provide insights into the impact of our method, rather than to conduct an exhaustive case study across various datasets. We believe that including additional datasets may introduce unnecessary complexity and potentially dilute the clarity of our findings.
> >
> > I agree that conducting exhaustive case studies is not meaningful in most cases, but `achieving performance comparable to models with significantly larger embedding dimensions` is considered one of the core contributions of this work, as highlighted at the end of Introduction. Relying on observations from a single dataset is insufficient to substantiate this claim.

---

> ### Author Response · Authors · 2025-02-28
>
> Dear Reviewer 2EYK,
>
> Thank you for your feedback. We have requested an extension of the discussion period until March 4th to accommodate your request. Below are our responses. We hope they address the reviewer’s remaining concerns.
>
> $\textbf{Connections to Y20 and the Position of Sec. 3.1 in the Paper}$
>
> The optimal representation of machine learning models is fundamental to achieving high performance, and significant efforts have been dedicated to developing methods that enhance model representation. In unsupervised learning, approaches such as BERT [1], CPC [2], SimCLR [3], and Masked Autoencoders [4] have been widely explored. Mixup can also be categorized as a method for optimizing representations within the empirical risk minimization framework (see Sec. 2 of the Mixup paper).
>
> Y20 theoretically analyzed the impact of negative sampling on graph node representation and derived the expression of the empirical realization of the covariance measure (Eq. 4 in our paper), leading to a condition for the negative sample distribution that enables near-optimal node embedding (Sec. 3.3 in Y20). In Sec. 3.1, we extend this expression to knowledge graph embedding (Theorem 3.1). Sec. 3.2 then introduces the EMU formulation, while Sec. 3.3 provides a theoretical analysis of the negative sample distribution in EMU. Specifically, by substituting EMU’s negative sample distribution (Eq. 10) into Theorem 3.1, we demonstrate that EMU can lead to a near-optimal embedding, culminating in Theorem 3.5.
> For clarity, this logical progression is briefly summarized at the beginning of Sec. 3.
>
> [1] Devlin, Jacob, et al. "Bert: Pre-training of deep bidirectional transformers for language understanding." Proceedings of the 2019 conference of the North American chapter of the association for computational linguistics: human language technologies, volume 1 (long and short papers). 2019.
>
> [2] Henaff, Olivier. "Data-efficient image recognition with contrastive predictive coding." International conference on machine learning. PMLR, 2020.
>
> [3] Chen, Ting, et al. "A simple framework for contrastive learning of visual representations." International conference on machine learning. PmLR, 2020.
>
> [4] He, Kaiming, et al. "Masked autoencoders are scalable vision learners." Proceedings of the IEEE/CVF conference on computer vision and pattern recognition. 2022.
>
> $\textbf{Additional Experiments on Scaling Dependence Using YAGO3-10}$
>
> Thank you for your constructive feedback. As suggested, we conducted additional experiments on YAGO3-10 using DistMult and HAKE, and the results are as follows:
>
> $$\\begin{array} {|r|r|}\\hline Dataset & Model & Case & (d, n) & MRR & HITS@10 \\\ \\hline YAGO3-10 & DistMult & w/t EMU & (100, 256) & 0.345 & 0.538 \\\ \\hline  &  & w/t EMU & (1000, 256) & 0.392 & 0.594 \\\ \\hline  &  & EMU & (100, 256) & 0.403 & 0.601 \\\ \\hline & HAKE & w/t EMU & (500, 256) & 0.452 & 0.651 \\\ \\hline  &  & w/t EMU & (1000, 256) & 0.482 & 0.665 \\\ \\hline  &  & EMU & (500, 256) & 0.499 & 0.687 \\\ \\hline \\end{array}$$
>
> As shown, increasing model size continues to improve performance even on YAGO3-10. Furthermore, DistMult with EMU achieves performance comparable to a model with 10 times the embedding dimension. These results have been incorporated into the updated version of the paper (Table 1).

---

> > ### Comment · Reviewer_2EYK · 2025-03-01
> >
> > Thank you for the response and additional experiments. My concerns have been addressed now. btw there is a typo in the numerator of eq.11

---

> > > ### Author Response · Authors · 2025-03-03
> > >
> > > Thank you for your supportive encouragement. We are pleased to inform you that the results of the HAKE experiment for the embedding dimension efficacy study (Section 4.3) have been obtained and are presented both above and in the updated manuscript (Table 1).
> > >
> > > > btw there is a typo in the numerator of eq.11.
> > >
> > > We sincerely appreciate your careful review. We believe the typo you referred to is the variable "u" on the left-hand side of the equation, which we have corrected to "z^t" in accordance with Equation 5 in the latest version of the manuscript.

---

> ### Comment · Action_Editor_Q6qJ · 2025-04-10
>
> Any remained comments for the revision?
>
> AE

---

### Author Response · Authors · 2025-02-19
**General Reply**

Dear Reviewers,

We sincerely appreciate your thoughtful and encouraging feedback. Following your suggestions, we have revised our paper accordingly, and the updated version is now available on OpenReview. The modified sections are highlighted in red. In this general response, we summarize our answers to common concerns raised by the majority of the reviewers.

$\textbf{Application of EMU to One of the Recent SOTA Models: NBFNet.}$

In response to the suggestions from Reviewer wpzR and Reviewer VhgB, we have applied EMU to Neural Bellman-Ford Networks (NBFNet), a state-of-the-art (SOTA) model on the FB15k-237 dataset. The experimental results are as follows:
$$\\begin{array} {|r|r|}\\hline  & MRR & hit@1 & hit@3 & hit@10 \\\ \\hline vanilla (reported) & 0.415 & 0.321 & 0.454 & 0.599 \\\ \\hline vanilla (our \ experiment) & 0.414 \\pm 0.003 & 0.321 \\pm 0.003 & 0.453 \\pm 0.003 & 0.596 \\pm 0.002 \\\ \\hline w.t. EMU & {\\bf 0.419 \\pm 0.002} & {\\bf 0.326 \\pm 0.002} & {\\bf 0.460 \\pm 0.003} & {\\bf 0.601 \\pm 0.002} \\\\ \hline  \end{array}$$

These results demonstrate that EMU remains effective even when applied to the latest models, achieving possibly a new SOTA performance, at least among NBFNet, A*Net, Adaprop, and PPR. While the performance gain may appear smaller compared to other models in our study, we attribute this to the additional structure in NBFNet—specifically, the Bellman-Ford Iteration—which differs from the architectures used in our other experiments. Further hyperparameter tuning could potentially enhance EMU’s performance on NBFNet, but due to time constraints during the rebuttal period, we were unable to conduct extensive tuning.

$\textbf{Possibility of Extension of the Theoretical Analysis (Loss Functions and Decoder Models)}$

In our study, we primarily applied EMU with the (binary) cross-entropy loss function. However, in theory, EMU is also compatible with other loss functions commonly used in link prediction tasks, such as margin loss and triplet loss, due to its inherent flexibility, though we did not conduct experiments with these alternative loss functions. This is because (binary) cross-entropy loss is widely recognized as the most effective loss function for knowledge graph embedding (KGE) tasks. This claim is supported by previous work (Ruffinelli et al., 2020, "You Can Teach an Old Dog New Tricks! On Training Knowledge Graph Embeddings", ICLR 2020), as well as by the fact that NBFNet itself employs binary cross-entropy loss to achieve its SOTA performance.
Regarding our theoretical analysis and the assumption of DistMult, we have revisited our proof and found that it does not rely explicitly on the DistMult formulation. Instead, our analysis applies to a more general function of the form f(u,v), where f is a scalar function that takes u and v as inputs, such as DistMult and neural network-based scoring functions. We have revised our discussion accordingly to reflect this broader applicability.

Once again, we truly appreciate your valuable feedback, which has helped improve our work. We look forward to your further insights.

Best regards,

Authors

---

### Decision · Action_Editor_Q6qJ · 2025-03-16

**Recommendation:** Accept with minor revision

**Comment:**

This paper introduces the Embedding Mutation (EMU) framework for generating negative samples in knowledge graph embedding. By extending the Y20 framework, the authors derive conditions for optimal negative sample distributions and demonstrate that mutating embedding vectors with EMU markedly improves link prediction performance across multiple datasets and models, achieving results comparable to models with fivefold larger embeddings.

Strength
- The paper offers a robust theoretical foundation with rigorous derivations that establish sufficient conditions for effective negative sample distributions.
- EMU introduces a novel mechanism for generating negative samples through embedding mutation, which contrasts with traditional methods that simply corrupt positive triples.
- The simplicity of EMU and its seamless integration with existing KGE models enhance its practical applicability.
- Extensive experiments across diverse datasets and models consistently demonstrate significant improvements in link prediction
performance.


Weakness:
- EMU demonstrates diminished performance on WN18RR, likely due to intrinsic latent space isotropy; further analysis could better elucidate this phenomenon.
- The mutation ratio (α) requires careful tuning, and the lack of adaptive parameter adjustment strategies may limit EMU’s robustness in diverse scenarios.

Please revise the submission accordingly, and ***add color to the revised version to make easy check.***

**Audience:**

This paper is highly relevant to the TMLR audience as it addresses the challenge of negative sampling in knowledge graph completion through a theoretically grounded and practically applicable EMU framework that is compatible with diverse KGE models.

**Claims And Evidence:**

Yes. This paper rigorously derives conditions for near-optimal knowledge graph embeddings by extending the theoretical framework of Y20 to the KG domain, demonstrating that generating negative samples through embedding mutation satisfies sufficient criteria for optimality. Empirical validation across multiple benchmark datasets and models reveals consistent performance improvements, with ablation studies and PCA visualizations confirming that EMU enables low-dimensional models to match the performance of embeddings five times larger. To address generalizability concerns, the authors conducted additional experiments on YAGO3-10 and expanded their theoretical analysis, reinforcing the method’s robustness. While reviewers have noted similarities to the Mixup data augmentation technique, the authors support these claims by highlighting EMU’s unique mechanism of generating hard negative samples and by providing superior empirical evidence.

---

> ### Author Response · Authors · 2025-04-01
> **response**
>
> Dear Action Editor,
>
> Thank you for your constructive feedback and for the opportunity to further improve our paper. Below, we outline the modifications made in the revised manuscript in response to your comments:
>
> W1: We have added a discussion in Footnote 5 on the effectiveness of EMU on WN18RR to better support our approach on this dataset.
>
> W2: We have included guidance on selecting the hyper-parameter of EMU in the final section of Appendix K, titled “Hyper-Parameter Dependence Study.”
>
> The modified and newly added sections are highlighted in red for clarity. Upon acceptance, we will prepare the unanonymized camera-ready version.
> We hope these revisions sufficiently address your concerns and that our paper is now suitable for acceptance.
>
> Best regards,
>
> Authors